# IBCL: Zero-shot Model Generation under Stability-Plasticity Trade-offs

## Abstract

Algorithms that balance the stability-plasticity trade-off are well studied in the Continual Learning literature. However, only a few focus on obtaining models for specified trade-off preferences. When solving the problem of continual learning under specific trade-offs (CLuST), state-of-the-art techniques leverage rehearsal-based learning, which requires retraining when a model corresponding to a new trade-off preference is requested. This is inefficient, since there potentially exists a significant number of different trade-offs, and a large number of models may be requested. As a response, we propose Imprecise Bayesian Continual Learning (IBCL), an algorithm that tackles CLuST efficiently. IBCL replaces retraining with a constant-time convex combination. Given a new task, IBCL (1) updates the knowledge base as a convex hull of model parameter distributions, and (2) generates one Pareto-optimal model per given trade-off via convex combination without additional training. That is, obtaining models corresponding to specified trade-offs via IBCL is zero-shot. Experiments whose baselines are current CLuST algorithms show that IBCL improves classification by at most 44% on average per task accuracy, and by 45% on peak per task accuracy while maintaining a near-zero to positive backward transfer, with memory overheads converging to constants. In addition, its training overhead, measured by the number of batch updates, remains constant at every task, regardless of the number of preferences requested. IBCL also improves multi-objective reinforcement learning tasks by maintaining the same Pareto front hypervolume, while significantly reducing the training cost. Details can be found at: `https://github.com/ibcl-anon/ibcl`.

## 1 Introduction

Continual Learning (CL), also known as lifelong machine learning, is a special case of multi-task learning, where tasks arrive in temporal sequence one-by-one (Thrun, 1998; Ruvolo & Eaton, 2013; Chen & Liu, 2016; Parisi et al., 2019). Two key properties matter for CL algorithms: stability and plasticity (De Lange et al., 2021). Here, stability means the ability to maintain performance on previous tasks, not forgetting what the model has learned, and plasticity refers to the ability to adapt to a new task. Unfortunately, these two properties are conflicting due to the multi-objective optimization nature of CL (Kendall et al., 2018; Sener & Koltun, 2018). For years, researchers have been balancing the stability-plasticity trade-off. However, few have discussed the problem of learning models for specifically given trade-off points. In this paper, we focus on such a problem, which we denote as CL under specific trade-offs (CLuST).

Why is CLuST important? First, in certain scenarios, it is important to explicitly specify *how much* stability and plasticity are needed to obtain a customized model for each trade-off preference. Second, when there exists a large number of preferences, the training *efficiency* of every customized model matters. Otherwise, the training cost accumulates on all preferences and becomes prohibitive. Therefore, we are not only looking for a solution to the CLuST problem, but also an efficient one.

**Motivating Example.** Consider an example of a movie recommendation system. The model is first trained to rate movies in the sci-fi genre. Then, the movie company adds a new genre, e.g., documentaries. The model needs to learn how to rate documentaries while not forgetting how to rate sci-fis. Training this model

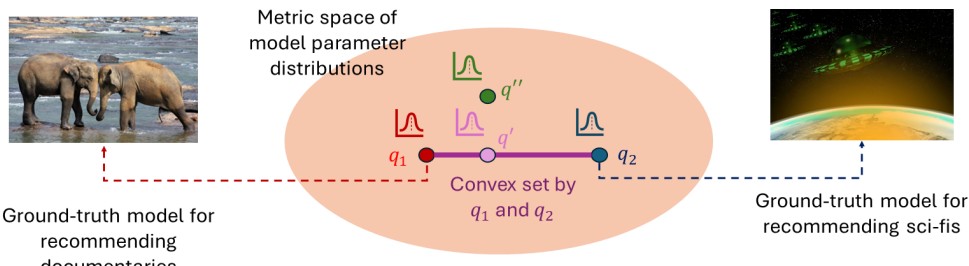

Figure 1: A Bayesian view of a Pareto-optimal parameter distribution $q'$ and a non-Pareto-optimal parameter distribution $q''$.

boils down to a CL problem. The company now wants to build a recommendation system that adapts to users' tastes in movies. For example, Alice has equal preferences over sci-fis and documentaries. Bob, however, wants to watch only documentaries and has no interest in sci-fis at all. Consequently, the company aims to train two customized models for Alice and Bob, respectively, to predict how likely a sci-fi or a documentary are to be recommended. Based on individual preferences, Alice's personal model should balance between the accuracy in rating sci-fis and rating documentaries, while Bob's model allows for compromising on the accuracy in rating sci-fis to achieve a high accuracy in rating documentaries. As new genres are added, users should be able to input their preferences on all available genres to obtain customized models. Since there could be many different users, and each user's taste in movie genres could vary over time, the movie company should implement models that adapt to *a significant number of preferences*. The costs would be prohibitive if the company had to train one model per distinct preference.

**The CLuST Problem and its Challenges.** To formalize the CLuST problem, we take a Bayesian perspective, where learnable model parameters are viewed as random variables (Farquhar & Gal, 2019; Kessler et al., 2023; Nguyen et al., 2018). As illustrated in Figure 1, we consider all parameter distributions living in a metric space. This metric can be any valid metric for distributions, such as the 2-Wasserstein distance (Deza & Deza, 2013). The figure shows an example of two sequential tasks, with ground-truth parameter distributions $q_1$ and $q_2$, respectively. From this setup, a distribution that emphasizes stability (in task 2) is a distribution closer to $q_1$ than $q_2$, and a distribution that prioritizes plasticity is closer to $q_2$ than $q_1$. Notice that irrespective of the desired stability-plasticity trade-off, we want the distribution to be *Pareto-optimal*, which loosely means that there is no way to improve such distribution by making it closer to *both* $q_1$ and $q_2$. We can see that Pareto-optimality is equivalent to being inside the convex set enclosed by $q_1$ and $q_2$. For example, $q'$ in the figure is a Pareto-optimal distribution, while $q''$ is not. With this setting, we can specify a trade-off point using a *preference vector* (Mahapatra & Rajan, 2020; 2021) $\bar{w} = (w_1, w_2)$, where $w_1, w_2 \geq 0$ and $w_1 + w_2 = 1$. The preferred Pareto-optimal distribution is, therefore, a convex combination $w_1 q_1 + w_2 q_2$.

So far, researchers have already proposed the use of preference vectors to specify trade-off points in multi-task and continual learning (Gupta et al., 2021; Lin et al., 2019; 2020; Ma et al., 2020). However, instead of using them as coefficients of distributions, state-of-the-art techniques use them as coefficients for loss functions in *rehearsal-based methods*. That is, existing algorithms memorize some data $d_i$ for each task $i$ (for "rehearsal"), and let the loss at task $i$ be $l_i = \sum_{j=1}^{i} w_j l(d_j)$, with $l$ being a generic loss function like cross-entropy. There are at least two drawbacks to this approach. First, rehearsals must retrain the entire model whenever we have a new trade-off preference. **In plain words, these methods have a training overhead proportional to the number of preferences at each task.** As numerous possible preferences exist, this boils down to an efficiency issue when there is a large number of preferences, such as many users in the movie recommendation example. It would be desirable if we could obtain the preferred models using training-free constant-time operations instead of retraining. Moreover, rehearsals must cache data, and stable performance on previous tasks depends on which data can be memorized.

**The IBCL Algorithm.** To overcome these shortcomings faced by CLuST algorithms, we propose Imprecise Bayesian Continual Learning (IBCL), whose workflow is illustrated in Figure 2. In step 1, upon arrival of

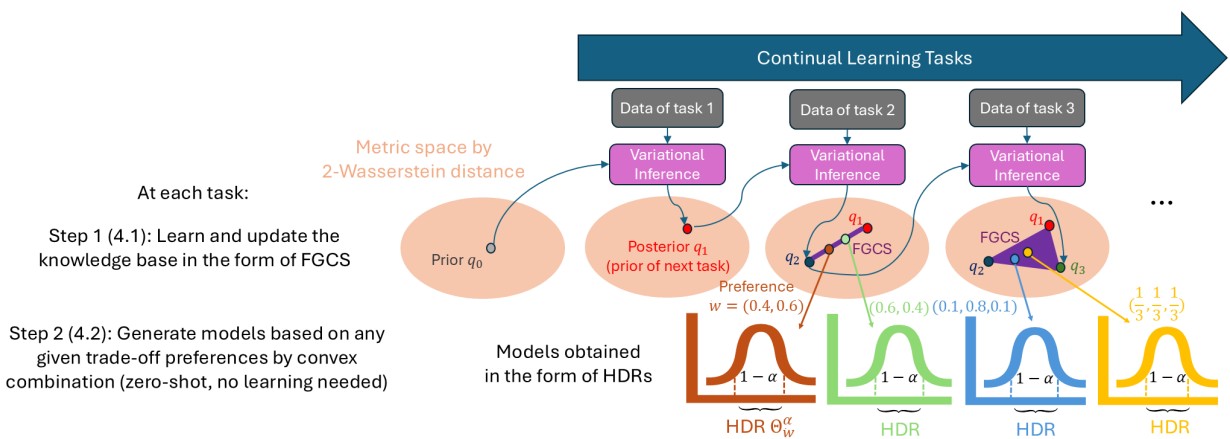

Figure 2: The workflow of Imprecise Bayesian Continual Learning (IBCL). Here, we start from 1 prior, but in practice, there may be more than 1 to reduce epistemic uncertainty (Hüllermeier & Waegeman, 2021).

the training data for a new task, IBCL updates its *knowledge base* (that is, all information shared across tasks) in the form of a convex set of distributions with finitely many extreme elements (the elements that cannot be written as convex combinations of one another), called *finitely generated credal set* (FGCS) (Caprio et al., 2024). This is done by variational inference from the learned distribution of the previous task, and the learned distributions serve as extreme elements of the FGCS. Each point in the FGCS corresponds to one Pareto-optimal distribution on the trade-off polytope of all tasks so far. Then, at step 2, given any preference vector $\bar{w}$, IBCL selects the preferred distribution by convex combination. A parameter region is obtained as a highest density region (HDR) of the selected distribution, which is the smallest parameter set that contains the ground-truth model with high probability.

IBCL addresses the identified shortcomings as follows. **First, IBCL replaces retraining in state-of-the-art with constant-time, zero-shot convex combination to generate models. It has a constant training overhead per task (to update the FGCS), independent of the number of preferences.** Additionally, no data cache is required, and therefore the stability of our model does not depend on the data memorized. Experiments on image classification and NLP benchmarks support the effectiveness of IBCL. We find that IBCL improves on the baselines by at most 44% in average per task accuracy, and by 45% in peak per task accuracy, while maintaining a near-zero to positive backward transfer, with a constant training overhead regardless of the number of preferences. Most importantly, IBCL has significantly smaller training time, costing only 6.3% to 9.6% of rehearsal-based baselines, measured in number of batch updates. We also show that IBCL has a sublinear memory growth along the number of tasks.

**Contributions.** In general, we have the following contributions:

1. We are the first to formulate the problem of Continual Learning under Specific Trade-offs (CLuST) from a Bayesian perspective. This problem requests one model per trade-off preference, and therefore demands efficiency due to a potentially large number of preferences (Section 3).

2. We propose Imprecise Bayesian Continual Learning (IBCL), a Bayesian CL algorithm that solves CLuST. IBCL leverages data structures from Imprecise Probability, and therefore is able to generate models to address arbitrary number of preferences at each task with a fixed training cost (Section 4).

3. We experiment IBCL on standard image classification and NLP CL benchmarks, with at most 44% improvement in average per task accuracy, 45% in peak per task accuracy, almost zero catastrophic forgetting, memory overhead converging to constants, and most importantly, the training time is significantly decreased to only 6.3% to 9.6% of baselines, measured in number of batch updates required (Section 5).

4. We also made an attempt to adapt IBCL to reinforcement learning tasks, resulting in the same level of Pareto front hypervolumes, while significantly reducing the training cost.

## 2 Background

### 2.1 Imprecise Probability

Our algorithm hinges upon the concepts of finitely generated credal set (FGCS) from Imprecise Probability (IP) theory (Walley, 1991; Augustin et al., 2014; Troffaes & de Cooman, 2014).[1]

**Definition 2.1** (Finitely Generated Credal Set). Given a finite set of probability distributions $\{q^j\}_{j=1}^m$, a finitely generated credal set (FGCS) is the convex set

$$\mathcal{Q} = \left\{ q : q = \sum_{j=1}^m \beta^j q^j, \ \beta^j \geq 0 \ \forall j \ , \ \sum_{j=1}^m \beta^j = 1 \right\}. \tag{1}$$

In other words, FGCS $\mathcal{Q}$ is the convex hull $\mathrm{CH}(\{q^j\}_{j=1}^m)$ of finitely many distributions $\{q^j\}_{j=1}^m$. That is, given a finite collection of distributions $\{q^j\}_{j=1}^m$ (that we call the extreme elements of the credal sets, and denote by $\mathrm{ex}[\mathcal{Q}]$), $\mathcal{Q}$ is the collection of all probability distributions $q$ that can be written as a convex combination of the $q^j$'s. If the state space is finite, then the $q^j$'s can be seen as probability vectors, whose entries represent the probability mass assigned by distribution $q^j$ to the elements of the state space. Working with sets of probabilities allows to mitigate problems ensuing from distribution misspecification and/or drift (Kaur et al., 2023; Lin et al., 2024).

Next, we borrow the idea of highest density region (HDR) (Coolen, 1992).

**Definition 2.2** (Highest Density Region). Let $\theta \in \Theta$ be a continuous random variable following a probability density function (pdf) $q$, with $\Theta$ being a set of interest.[2] Given a significance level $\alpha \in [0, 1]$, the $(1-\alpha)$-HDR is a subset $\Theta_q^\alpha \subset \Theta$, such that

$$\int_{\Theta_q^\alpha} q(\theta) d\theta \geq 1 - \alpha, \ \text{ and } \ \int_{\Theta_q^\alpha} d\theta \text{ is minimal}. \tag{2}$$

In equation 2, requiring that $\int_{\Theta_q^\alpha} d\theta$ is minimal corresponds to requiring that $\Theta_q^\alpha$ has the smallest possible cardinality (i.e., the least possible number of elements). Indeed, if $\Theta$ is finite, it can be replaced by "$|\Theta_q^\alpha|$ is minimal". Since we consider the most general case (in which set $\Theta$ may be uncountable), we must use the integral notion instead of cardinality, as pointed out in previous research (Coolen, 1992). In turn, equation 2 tells us that $\Theta_q^\alpha$ is the set having the smallest number of elements that also satisfies $\mathrm{Pr}_{\theta \sim q}[\theta \in \Theta_q^\alpha] \geq 1 - \alpha$, provided that $\theta \sim q$.

To further explain HDR, an equivalent definition is as follows (Hyndman, 1996).

**Definition 2.3** (Highest Density Region, Alternative). Let $\Theta$ be a set of interest, and consider a significance level $\alpha \in [0, 1]$. Suppose that a (continuous) random variable $\theta \in \Theta$ has probability density function (pdf) $q$.

The $\alpha$-*level Highest Density Region (HDR)* $\Theta_q^\alpha$ is the subset of $\Theta$ such that

$$\Theta_q^\alpha = \{\theta \in \Theta : q(\theta) \geq q^\alpha\}, \tag{3}$$

where $q^\alpha$ is a constant value. In particular, $q^\alpha$ is the largest constant such that $\mathrm{Pr}_{\theta \sim q}[\theta \in \Theta_q^\alpha] \geq 1 - \alpha$.

Some scholars indicate HDRs as the Bayesian counterpart to the frequentist concept of confidence intervals. In dimension 1, $\Theta_q^\alpha$ can be interpreted as the narrowest interval – or union of intervals – in which the value of the (true) parameter falls with probability of at least $1 - \alpha$ according to distribution $q$. We give a simple visual example in Figure 3.

---

[1]For more modern references, see e.g. Caprio & Mukherjee (2023); Caprio & Seidenfeld (2023); Caprio et al. (2025); Dutta et al. (2025); Caprio (2025); Chau et al. (2025); Sloman et al. (2025).

[2]Here, for ease of notation, we do not distinguish between a random variable and its realization.

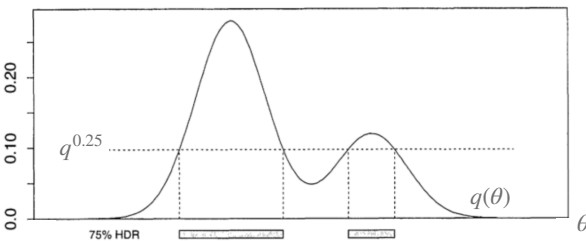

Figure 3: The 0.25-HDR for a Normal Mixture density. This is a replica of Hyndman (1996, Figure 1).

## 2.2 Continual Learning

Continual Learning, also known as lifelong learning, is a special case of multitask learning, where tasks arrive sequentially rather than simultaneously (Thrun, 1998; Ruvolo & Eaton, 2013). In this paper, we leverage Bayesian inference in the knowledge base update (Ebrahimi et al., 2019). Like generic multitask learning, continual learning also faces the stability-plasticity trade-off (De Lange et al., 2021), which balances between performance on new tasks and resistance to catastrophic forgetting (Kirkpatrick et al., 2017). Current methods identify models to address trade-off preferences by techniques such as loss regularization (Servia-Rodriguez et al., 2021), which means that at least one model must be trained per preference.

Researchers in CL have proposed various approaches to retain knowledge while updating a model on new tasks. These include modified loss landscapes for optimization (Farajtabar et al., 2020), preservation of critical pathways via attention (Abati et al., 2020), memory-based methods (Lopez-Paz & Ranzato, 2017), shared representations (Lee et al., 2019), and dynamic representations (Bulat et al., 2020).

One sub-category of CL is Bayesian Continual learning (BCL), which leverages probabilistic methods, defined as follows (Nguyen et al., 2018).

**Definition 2.4** (Bayesian Continual Learning)**.** BCL is a class of CL procedures, which starts with a prior distribution $q_0$. At a task $i \in \{1, 2, ...\}$, we are given i.i.d. training data $\{(x_s, y_s)\}_{s=1}^{n_i} \subset \mathcal{X} \times \mathcal{Y}$ of inputs and outputs (the $x$'s and $y$'s, respectively). The Bayesian model is updated from prior distribution $q_{i-1}$ to posterior $q_i$ using the labeled data. Then, $q_i$ is used as a prior in task $i + 1$.

In BCL (Nguyen et al., 2018; Ebrahimi et al., 2019), each task is associated with a data generating process parameterized by $\theta$. The latter is postulated to be a random quantity, which at the beginning of the analysis has a prior distribution, $\theta \sim q_0$. After training on the available data, the prior distribution is turned into posterior, $\theta \sim q_1$ via Bayes' theorem. The posterior $q_1$ is the revised parameter distribution after having learned from the data pertaining to the first task to complete. The posterior is then used as a prior for the next task.

A significant application of CL is continual reinforcement learning (CRL). In reinforcement learning (Li, 2017), an agent learns an optimal control policy to maximize a cumulative reward in an environment. Formally, the system contains a state space $\mathcal{S}$ and an action space $\mathcal{A}$, with an underlying Markov decision process (MDP) that characterizes the state transition upon an action taken. The reward is a real-valued function $r : \mathcal{S} \times \mathcal{A} \times \mathcal{S} \to \mathbb{R}$ that maps from the previous state, the action, and the next state to a score. When the MDP is non-stationary, researchers have proposed CRL algorithms to solve for optimal policies (Khetarpal et al., 2022). The stationary nature could be broken due to different drivers, including active changes of the agent, passive changes in the environment, and a hybrid of both.

## 2.3 Stability-plasticity Trade-offs

Researchers in multitask and continual learning have explored the trade-off nature among tasks. The reason behind this trade-off is due to using shared parameters for multiple tasks. Therefore, optimizing the parameters would be a multi-objective optimization problem, which would lead to a non-singleton Pareto set of parameters (Caruana, 1997; Sener & Koltun, 2018).

In the context of continual learning, task performance trade-offs are formally known as stability-plasticity trade-offs, a terminology first introduced in 2013 (Mermillod et al., 2013) and also known as the stability-plasticity dilemma. Here, stability means the ability to maintain performance in previously encountered tasks, and plasticity means the ability to obtain performance in new tasks. Inspired by biological neural systems, this trade-off describes a continuum of catastrophic forgetting. Specifically, a learned representation has high stability (and low stability) if its learned internal representations of distinct objectives have small overlaps in parameters. Then, a backpropagation that updates the parameters of one objective would have little effect on other objectives. However, it is impossible to completely segregate the parameters of objectives, as it would lead to an explosion in the total number of parameters required when learning more tasks. Consequently, catastrophic forgetting is almost inevitable given a limited number of parameters, and the stability-plasticity trade-off implies a Pareto front of parameter configurations, which is a continuous manifold.

Researchers have been developing continual learning algorithms with the awareness of this trade-off. For instance, the objective of stability and plasticity can be separately learned by distinct sub-networks (Kim et al., 2023; Lu et al., 2025), and multiple knowledge bases can be used to balance between stability and plasticity (Mahmoodi et al., 2025). Moreover, meta learning is also used to capture the common knowledge across tasks, and use only a small overhead on top of the learned common knowledge to achieve task-specific parameters (Caccia et al., 2020; Chen et al., 2023). However, all these methods are inflexible when we demand models at specific trade-offs. For example, the primary model in meta learning does not have a representation of trade-off as input. We have a detailed discussion on meta learning in Appendix A.

## 2.4 Continual Learning under Specific Trade-offs (CLuST)

Although we introduced the term "CLuST", previous research have already discussed the relevant topic of obtaining Pareto-optimal models at particular trade-off preferences. We borrow the formalization of preferences from established literature (Mahapatra & Rajan, 2020), where a preference is defined as a vector of non-negative real weights $\bar{w}$, with each entry $w_i$ corresponding to task $i$. That is, $w_i \geq w_j \iff i \succeq j$. This means that if $w_i \geq w_j$, then task $i$ is preferred to task $j$. However, state-of-the-art algorithms require training one model per preference, imposing large overhead when there are a large number of preferences.

Specifically, a given preference can guide the learning algorithms to find a corresponding particular model in the Pareto set. In multitask classification, researchers first consider a fixed set of preferences, each induces a single constrained subproblem, and learn one model per subproblem in parallel (Lin et al., 2019). This method is then enhanced by learning a hypernetwork that takes preferences as input and outputs model parameters, so that Pareto-optimal solutions can be learned with dynamic sets of preferences (Lin et al., 2020). Alternative architectures are also used, such as a linear preference-conditioned layer to improve computational efficiency (Ma et al., 2020) and HNPF models (Gupta et al., 2021). In addition, the Bayesian version of hypernetworks, formally known as multi-objective Bayesian optimization (MOBO) has also been explored (Lin et al., 2022). This method aims to learn an entire Pareto set represented by a single distribution, with preference information being a part of the parameters. Although MOBO aims for comprehensive posterior knowledge, obtaining such knowledge could be impractical: First, MOBO assumes data at any preference point is available by preference-based sampling, and even if so, it requires training on data sampled at *all* preferences. Second, MOBO is unable to efficiently update the distribution when task data arrives sequentially (i.e. continual learning). When more data arrives, MOBO has to update the entire distribution with an enormous number of parameters.

Learning under specific preferences is less explored when tasks are in sequential temporal orders, i.e., CLuST. State-of-the-art CLuST methods leverage rehearsal-based algorithms. The demand of balancing between stability-plasticity trade-off is first identified by researchers in 2021 (Raghavan & Balaprakash, 2021). Then, preferences are used as regularization factors on rehearsal data in 2023 (Kim et al., 2023). That is, given a vector of preferences $\bar{w} = (w_1, \cdot, w_m)$ on $m$ tasks, its Pareto-optimal model is learned by a loss function $L = \sum_{i=1}^{m} L_i$, where $L_i$ is a given loss function on the cached rehearsal data of task $i$. This approach can be equipped with various advanced rehearsal-based CL algorithms, such as GEM (Lopez-Paz & Ranzato, 2017), A-GEM (Chaudhry et al., 2018), DER, and DER++ (Buzzega et al., 2020). We also identify the major drawback of using rehearsal-based approaches in CLuST: they have to *train* the model for every preference, causing inefficiency when there are a large number of preferences to be addressed. Instead of rehearsal-based

methods, we propose to efficiently *generate* the majority of Pareto-optimal models at different preferences from only a few models trained.

In multitask and continual reinforcement learning, starting from 2024, researchers also designed methods for learning Pareto sets at different preferences. For instance, multi-objective gradients can be balanced with preference weights (Xu et al., 2020), and Bellman equations can be combined with preferences (Basaklar et al., 2022). Hypernetworks are also utilized. For example, Hyper-MORL learns a mapping from preferences to Pareto-optimal control policies using a hypernetwork representation (Shu et al., 2024). An alternative approach is to train the hypernetwork that maps to only a subset of parameters, which will then be appended to the policy parameters learned independently (Liu et al., 2025).

## 3 Formulating the CLuST Problem

In this section we formalize the CLuST problem. We consider domain-incremental learning (Van de Ven & Tolias, 2019; Shi & Wang, 2023) for classification models, with an unbounded number of stability-plasticity trade-off preferences at each task. The goal is to construct a learning algorithm with training overhead independent of the number of preferences, and that enjoys performance guarantees.

### 3.1 Assumptions

Let $\mathcal{X}$ be the space of inputs, and $\mathcal{Y}$ be the space of labels. In a typical classification problem, $\mathcal{X}$ will be a subset of a Euclidean space, and $\mathcal{Y}$ a finite set. In a typical regression problem, $\mathcal{Y}$ will too be a subset of a Euclidean space. In general, we do not limit ourselves to either scenario. As a consequence, we let the input and the output spaces be generic sets. Call $\Delta_{\mathcal{X}\mathcal{Y}}$ the space of all possible distributions on $\mathcal{X} \times \mathcal{Y}$. A task $i$ is associated with a distribution $p_i \in \Delta_{\mathcal{X}\mathcal{Y}}$, from which labeled data can be drawn i.i.d.

A common assumption in CL is *task similarity*, which researchers formalize as closeness in data distributions (Wang et al., 2024). Here, we have the same assumption. To define task similarity, we first recall the concept of 2-Wasserstein metric (Deza & Deza, 2013) on the data distributions.

**Definition 3.1** (2-Wasserstein Metric on $\Delta_{\mathcal{X}\mathcal{Y}}$)**.** The 2-Wasserstein metric is a distance $||\cdot||_{W_2}$ that measures the dissimilarity between two probability distributions $p$ and $p' \in \Delta_{\mathcal{X}\mathcal{Y}}$, with

$$\|p - p'\|_{W_2} := \left( \inf_{\gamma \in \Gamma(p,p')} \mathbb{E}_{((x_1,y_1),(x_2,y_2))\sim\gamma}[d((x_1,y_1),(x_2,y_2))^2] \right)^{\frac{1}{2}}, \tag{4}$$

where

1. $\Gamma(p, p')$ is the set of all couplings of $p$ and $p'$. A coupling $\gamma$ is a joint probability measure on $(\mathcal{X}\times\mathcal{Y})\times(\mathcal{X}\times\mathcal{Y})$ whose marginals are $p$ and $p'$ on the first and second factors, respectively, and

2. $d$ is the product metric endowed to $\mathcal{X} \times \mathcal{Y}$.[3]

With Definition 3.1, we have the following assumption.

**Assumption 3.2** (Task Similarity)**.** *For all task $i$, $p_i \in \mathcal{F}$, where $\mathcal{F}$ is a convex subset of $\Delta_{\mathcal{X}\mathcal{Y}}$. Also, we assume that the diameter of $\mathcal{F}$ is some $r > 0$, that is, $\sup_{p,q\in\mathcal{F}} \|p - q\|_{W_2} \leq r$, where $\|\cdot\|_{W_2}$ denotes the 2-Wasserstein distance.*

Assumption 3.2 states that the true data-generating processes about different tasks are not too distant. In addition, such a notion of "being not too distant" is entirely in the hands of the user, via the choice of radius $r$ and of the metric to endow $\Delta_{\mathcal{X}\mathcal{Y}}$. This assumption means that we do not expect very dissimilar tasks. That is, we do not consider e.g. a situation in which a robot is able to fold our clothes (task 1) and then deliver a payload in combat zone (task 2). Details of this assumption, including its importance, and why 2-Wasserstein metric is chosen, are explained in Section 3.2.

Next, we assume the parameterization of class $\mathcal{F}$.

---

[3]We denote by $d_{\mathcal{X}}$ and $d_{\mathcal{Y}}$ the metrics endowed to $\mathcal{X}$ and $\mathcal{Y}$, respectively.

**Assumption 3.3** (Parameterization of Task Distributions). *Every distribution $F$ in $\mathcal{F}$ is parameterized by $\theta$, a parameter belonging to a parameter space $\Theta$.*

Let us give an example of a parameterized family $\mathcal{F}$. Suppose that we have one-dimensional data points and labels. At each task $i$, the marginal on $\mathcal{X}$ of $p_i$ is a Gaussian $\mathcal{N}(\mu, 1)$, while the conditional distribution of label $y \in \mathcal{Y}$ given data point $x \in \mathcal{X}$ is a categorical $\text{Cat}(\vartheta)$. Hence, the parameter for $p_i$ is $\theta = (\mu, \vartheta)$, and it belongs to $\Theta = \mathbb{R} \times \mathbb{R}^{|\mathcal{Y}|}$. In this situation, an example of a family $\mathcal{F}$ that satisfies Assumptions 3.2 and 3.3 is the convex hull of distributions that can be decomposed as we just described, and whose distance according to the 2-Wasserstein metric does not exceed some $r > 0$.

Notice that all tasks share the same input space $\mathcal{X}$ and label space $\mathcal{Y}$, and we do not have task id's as an additional input, so learning is domain-incremental (Van de Ven & Tolias, 2019).

Preferences over stability-plasticity trade-offs is also an established concept (Mahapatra & Rajan, 2020; Servia-Rodriguez et al., 2021). We formalize it as follows.

**Definition 3.4** (Stability-plasticity Trade-off Preferences over Tasks). *Consider $k$ tasks with underlying data distributions $p_1, p_2, \ldots, p_k$. We express a stability-plasticity trade-off preference (or simply, a preference) over them through a probability vector $\bar{w} = (w_1, w_2, \ldots, w_k)^\top$. That is, $w_i \geq 0$ for all $i \in \{1, \ldots, k\}$, and $\sum_{i=1}^{k} w_i = 1$.*

Based on Definition 3.4, given a preference $\bar{w}$ over all $k$ tasks encountered, the personalized model for the user aims to learn the distribution $p_{\bar{w}} := \sum_{i=1}^{k} w_i p_i$. Since $p_{\bar{w}}$ is the convex combination of $p_1, \ldots, p_k$, thanks to Assumptions 3.2 and 3.3, we have $p_{\bar{w}} \in \mathcal{F}$, and therefore it is also parameterized by some $\theta \in \Theta$.

Like in existing BCL literature (Nguyen et al., 2018; Kessler et al., 2023; Servia-Rodriguez et al., 2021), we assume that the learning procedure is Bayesian domain-incremental learning. That is, the learning follows BCL as in Definition 2.4, and all data and label distributions are similar, as per Assumption 3.2, without any knowledge of task id's. At any task $k$, we are given at least one user preference $\bar{w}$ over the $k$ tasks so far. The data drawn for task $k + 1$ will not be available until we have finished learning models for all preferences on task $k$.

Generally, domain-incremental learning is harder than task-incremental learning because the former uses strictly less information. Extension from domain-incremental to task-incremental learning is trivial. To achieve such an extension, we only need to provide task ids as an additional input at both training and testing time.

### 3.2 Details of Assumption 3.2

We need Assumption 3.2 in light of the results in Kessler et al. (2023), where it is shown that misspecified models can suffer from catastrophic forgetting even when Bayesian inference is carried out exactly. By requiring that $\text{diam}(\mathcal{F}) = r$, we control the amount of misspecification via $r$. In Kessler et al. (2023), the authors design a new approach – called Prototypical Bayesian Continual Learning, or ProtoCL – that allows dropping Assumption 3.2 while retaining the Bayesian benefit of remembering previous tasks. Because the main goal of this paper is to come up with a procedure that allows the designer to express preferences over the tasks, we retain Assumption 3.2, and we work in the classical framework of Bayesian Continual Learning. In the future, we plan to generalize our results by operating with ProtoCL.[4]

Generally, we need a distance metric on distributions (i.e., non-negative, symmetric, and following the triangular rule), and we are aware of alternative metrics such as the square root of Jensen-Shannon (JS) divergence (Fuglede & Topsoe, 2004). However, most of these metrics, including square root of JS divergence, do not have a closed-form expression on Gaussians, which are commonly used in Bayesian inference. We choose the 2-Wasserstein distance for the ease of computation as it has an efficient closed-form. In practice, when all distributions are modeled by Bayesian neural networks with independent Gaussian weights and biases, we have

$$\|q_1 - q_2\|_{W_2}^2 = \|\mu_{q_1}^2 - \mu_{q_2}^2\|_2^2 + \|\sigma_{q_1}^2 \mathbf{1} - \sigma_{q_2}^2 \mathbf{1}\|_2^2, \tag{5}$$

---

[4]In Kessler et al. (2023), the authors also show that if there is a task dataset imbalance, then the model can forget under certain assumptions. To avoid complications, in this work we tacitly assume that task datasets are balanced.

where $\| \cdot \|_2$ denotes the Euclidean norm, $\mathbf{1}$ is a vector of all 1's, and $\mu_q$ and $\sigma_q$ are respectively the mean and standard deviation of a multivariate normal distribution $q$ with independent dimensions, $q = \mathcal{N}(\mu_q, \sigma_q^2 I)$, $I$ being the identity matrix. Therefore, computing the $W_2$-distance between two distributions is equivalent to computing the difference between their means and variances.

### 3.3 Main Problem

We aim to design a domain-incremental learning algorithm that generates one model per preference over tasks, with an unbounded number of preferences over a finite number of tasks. Given a significance level $\alpha \in [0, 1]$, in any task $k$, the algorithm should satisfy:

1. **Zero-shot preferred model generation**. A fixed training cost is needed at each task, regardless of the number of preferences. In other words, we only need to train a small fixed number of models per task, and after that, model generation for any preference is zero-shot.

2. **Probabilistic Pareto-optimality**. Let $\hat{q}_{\bar{w}}$ denote the convex combination of the estimated parameter distributions for tasks $1, \ldots, k$ using preference weights $\bar{w}$. IBCL should be able to identify the smallest subset of model parameters, $\Theta_{\hat{q}_{\bar{w}}}^{\alpha} \subset \Theta$ (written as $\Theta_{\bar{w}}^{\alpha}$ for notational convenience from now on), that contains the Pareto-optimal parameter $\theta_{\bar{w}}^{\star}$ with high probability. Formally, $\Theta_{\bar{w}}^{\alpha}$ is the minimal set that satisfies $\Pr_{\theta_{\bar{w}}^{\star} \sim \hat{q}_{\bar{w}}}[\theta_{\bar{w}}^{\star} \in \Theta_{\bar{w}}^{\alpha}] \geq 1 - \alpha$.

3. **Sublinear buffer growth**. The memory overhead accumulated by IBCL throughout the tasks should grow sublinearly in the number of tasks.

## 4 Imprecise Bayesian Continual Learning

Figure 2 shows that IBCL performs two steps in each task. First, it updates the knowledge base as a FGCS (Section 4.1). Second, it uses a convex combination of the extreme elements of the FGCS, instead of retraining, to zero-shot generate models under given preferences (Section 4.2).

### 4.1 FGCS Knowledge Base Update

As discussed in the Introduction, we take a Bayesian Continual Learning (BCL) approach, that is, the parameter $\theta$ of the distribution $p_k$ related to task $k$ is viewed as a random variable distributed according to some distribution $q$.

At the beginning of the analysis, we specify $m$ many such distributions, $\text{ex}[\mathcal{Q}_0] = \{q_0^1, \ldots, q_0^m\}$. They are the ones that the designer deems plausible – a priori – for the parameter $\theta$ of the task 1. Upon observing data from task 1, we learn a set $\mathcal{Q}_1^{tmp}$ of posterior parameter distributions and buffer them as extreme elements $\text{ex}[\mathcal{Q}_1]$ of the FGCS $\mathcal{Q}_1$ corresponding to task 1. We proceed in a similar way for successive tasks $i \geq 2$.

In Algorithm 1, we use notation $(\bar{x}_i, \bar{y}_i)$ to denote vectors of inputs and outputs pertaining to task $i$. In task $i$, we approximate $m$ posteriors $q_i^1, \ldots q_i^m$ by variational inference from buffered priors $q_{i-1}^1, \ldots q_{i-1}^m$ one-by-one (line 3). Variational inference is a standard Bayesian learning procedure that minimizes the evidence lower bound (ELBO) loss to infer a posterior distribution from a prior and observed data (Nguyen et al., 2018). However, if a posterior is very similar to an existing prior in the cache, it would give estimations with negligible differences to that prior. In this case, buffering this new posterior would be a waste in space. Therefore, we use a distance threshold $d$ to exclude the posteriors that are similar to the distributions that are already buffered (lines 4 - 10). When distributions similar to $q_i^j$ (within threshold $d$) are found in the knowledge base, we store a pointer to the distribution with minimal distance in place of $q_i^j$, and do not memorize $q_i^j$ (lines 8-9). The posteriors that are sufficiently different from the already buffered distributions are then appended to the knowledge base (line 12).

Notice that the memory overhead of Algorithm 1 is remembering at most $m$ distributions into $\mathcal{Q}_i^{tmp}$ at line 6. In practice, $m$ is a small constant (we choose $m = 3$ in our experiments). Therefore, the memory complexity is $O(1)$. Moreover, some newly memorized distributions may be discarded and replaced by a

---

**Algorithm 1** FGCS Knowledge Base Update

---

1: **Input:** Current knowledge base in the form of FGCS extreme elements $\text{ex}[\mathcal{Q}_{i-1}] = \{q_{i-1}^1, \ldots, q_{i-1}^m\}$, observed labeled data $(\bar{x}_i, \bar{y}_i)$ at task $i$, and distribution distance threshold $d \geq 0$
2: **Output:** Updated extreme elements $\text{ex}[\mathcal{Q}_i]$
3: $\mathcal{Q}_i^{tmp} \leftarrow \emptyset$
4: **for** $j \in \{1, \ldots, m\}$ **do**
5:     $q_i^j \leftarrow \mathsf{variational\_inference}(q_{i-1}^j, \bar{x}_i, \bar{y}_i)$
6:     $d_i^j \leftarrow \min_{q \in \text{ex}[\mathcal{Q}_{i-1}]} \|q_i^j - q\|_{W_2}$
7:     **if** $d_i^j \geq d$ **then**
8:         $\mathcal{Q}_i^{tmp} \leftarrow \mathcal{Q}_i^{tmp} \cup \{q_i^j\}$                    {Store distribution $q_i^j$}
9:     **else**
10:         $q_i^j \leftarrow \arg\min_{q \in \text{ex}[\mathcal{Q}_{i-1}]} \|q_i^j - q\|_{W_2}$  {Fetch the stored distribution with minimal distance to $q_i^j$, and overwrite $q_i^j$ with a pointer to that distribution}
11:         $\mathcal{Q}_i^{tmp} \leftarrow \mathcal{Q}_i^{tmp} \cup \{q_i^j\}$                    {Only a pointer is stored}
12:     **end if**
13: **end for**
14: $\text{ex}[\mathcal{Q}_i] \leftarrow \text{ex}[\mathcal{Q}_{i-1}] \cup \mathcal{Q}_i^{tmp}$

---

previous distribution in cache at line 8. With larger threshold $d$ at line 5, more distributions are discarded at lines 8-9. The amortized memory complexity analysis under different threshold $d$'s is discussed in our ablation studies: see Section 5.2.

The time complexity of Algorithm 1 is dominated by variational inference at line 3. Every variational inference costs a non-negligible training time, which we denote as $O(v)$. There are a total of $m$ variational inferences computed for each task. Since $m$ is a constant, the overall time complexity remains $O(v)$.

### 4.2 Zero-shot Generation of User Preferred Models

Next, after having updated the FGCS extreme elements for task $i$, we are given a set of user preferences. For each preference $\bar{w}$, we need to identify the Pareto-optimal parameter $\theta_{\bar{w}}^\star$ for the preferred data distribution $p_{\bar{w}}$. This procedure can be divided into two steps as follows.

First, we find the parameter distribution $\hat{q}_{\bar{w}}$ via a convex combination of the extreme elements in the knowledge base, whose weights correspond to the entries of preference vector $\bar{w} = \{w_1, \ldots, w_i\}$ over the $i$ tasks so far. That is,

$$\hat{q}_{\bar{w}} = \sum_{k=1}^{i} \sum_{j=1}^{m_k} \beta_k^j q_k^j, \text{ where } \sum_{j=1}^{m_k} \beta_k^j = w_k, \text{ and } \beta_k^j \geq 0, \text{ for all } j \text{ and all } k. \tag{6}$$

Here, $q_k^j$ is a buffered extreme point of FGCS $\mathcal{Q}_k$, i.e. the $j$-th parameter posterior of task $k$. The weight $\beta_k^j$ of this extreme point is decided by preference vector entry $\bar{w}_j$. In implementation, if we have $m_k$ extreme elements stored for task $k$, we can choose equal weights $\beta_k^1 = \cdots = \beta_k^m = w_k/m_k$. For example, if we have preference $\bar{w} = (0.8, 0.2)^\top$ on two tasks so far, and we have two extreme elements per task stored in the knowledge base, we can use $\beta_1^1 = \beta_1^2 = 0.8/2 = 0.4$ and $\beta_2^1 = \beta_2^2 = 0.2/2 = 0.1$. In practice, we use $m_k = 1$ for all tasks. That is, we learn one parameter posterior $q_k$ at every task $k$, and therefore $\hat{q}_{\bar{w}} = \sum_{k=1}^{i} w_k q_k$. We also have ablation studies on different $m_k$. See Section 5 for details.

The following proposition ensures us that it is equivalent to express preferences over tasks $k$, or over the parameter distributions $q_k^j$ associated with each task, thus justifying the definition of $\hat{q}_{\bar{w}}$ in equation 6.

**Proposition 4.1** (Selection Equivalence). *Let $q_k^j$ be an extreme point posterior of $\mathcal{Q}_i$ learned from the $j$-th prior at task $k \in \{1, \ldots, i\}$. For any preference $\bar{w} = (w_1, \ldots, w_i)^\top$ on tasks $\{1, \ldots, i\}$, there exists a*

*probability vector $\bar{\beta} = (\beta_1^1, \ldots, \beta_1^{m_1}, \ldots, \beta_i^1, \ldots, \beta_i^{m_i})^\top$, with $\sum_{j=1}^{m_k} \beta_k^j = w_k$, for all $k \in \{1, \ldots, i\}$, such that*

$$\hat{q}_{\bar{w}} = \sum_{k=1}^{i} \sum_{j=1}^{m_k} \beta_k^j q_k^j.$$

*In other words, selecting a precise distribution $\hat{q}_{\bar{w}}$ from $\mathcal{Q}_i$ is equivalent to specifying a preference weight vector $\bar{w}$ on tasks $\{1, \ldots, i\}$.*

We refer to Appendix B for the proof.

Second, we compute the HDR $\Theta_{\bar{w}}^\alpha \subset \Theta$ from $\hat{q}_{\bar{w}}$. This is implemented using a standard procedure that locates the smallest region in the parameter space whose enclosed probability mass is (at least) $1 - \alpha$, according to $\hat{q}_{\bar{w}}$. This procedure can be routinely implemented, e.g., in R, using package HDInterval (Juat et al., 2022). As a result, we locate the smallest set of parameters $\Theta_{\bar{w}}^\alpha \subset \Theta$ associated with the preference $\bar{w}$. This subroutine is formalized in Algorithm 2. Notice that this computation is simply a convex combination, i.e., a weighted sum of all distributions in $\text{ex}[\mathcal{Q}_i]$. The summation is defined under 2-Wasserstein metric. As explained in Section 3.2, the convex combination has a computational complexity proportional to the parameterization size of distributions. In practice, we first extract features from the data and use a relatively small parameterization for distributions on top of the extracted features. Please see Section 5 for details. Furthermore, this algorithm does not produce any memory overhead.

---

**Algorithm 2** Preference HDR Computation

1: **Input:** Knowledge base $\text{ex}[\mathcal{Q}_i]$ with $m_k$ extreme elements saved for task $k \in \{1, \ldots, i\}$, preference $\bar{w}$ on the $i$ tasks, significance level $\alpha \in [0, 1]$
2: **Output:** HDR $\Theta_{\bar{w}}^\alpha \subset \Theta$
3: **for** $k = 1, \ldots, i$ **do**
4:    $\beta_k^1 = \cdots = \beta_k^m \leftarrow w_k / m_k$
5: **end for**
6: $\hat{q}_{\bar{w}} = \sum_{k=1}^{i} \sum_{j=1}^{m_k} \beta_k^j q_k^j$
7: $\Theta_{\bar{w}}^\alpha \leftarrow \text{hdr}(\hat{q}_{\bar{w}}, \alpha)$

---

### 4.3 Overall IBCL Algorithm and Analysis

From the two subroutines in Sections 4.1 and 4.2, we construct the overall IBCL algorithm as in Algorithm 3.

---

**Algorithm 3** Imprecise Bayesian Continual Learning

1: **Input:** Prior distributions $\text{ex}[\mathcal{Q}_0] = \{q_0^1, \ldots, q_0^m\}$, hyperparameters $\alpha$ and $d$
2: **Output:** HDR $\Theta_{\bar{w}}^\alpha$ for each given preference $\bar{w}$ at each task $i$
3: **for** task $i = 1, 2, \ldots$ **do**
4:    $\bar{x}_i, \bar{y}_i \leftarrow$ sample $n_i$ labeled data points i.i.d. from $p_i$
5:    $\text{ex}[\mathcal{Q}_i] \leftarrow \text{fgcs\_update}(\text{ex}[\mathcal{Q}_{i-1}], \bar{x}_i, \bar{y}_i, d)$                    {Algorithm 1}
6:    **while** user has a new preference **do**
7:       $\bar{w} \leftarrow$ user input
8:       $\Theta_{\bar{w}}^\alpha \leftarrow \text{preference\_hdr\_computation}(\text{ex}[\mathcal{Q}_i], \bar{w}, \alpha)$                    {Algorithm 2}
9:    **end while**
10: **end for**

---

For each task, in line 3, we use Algorithm 1 to update the knowledge base by learning $m$ posteriors from the current priors. In lines 5-7, according to a user-given preference over all tasks so far, we obtain the HDR of the model associated with preference $\bar{w}$ in zero-shot via Algorithm 2. Notice that this HDR computation does not require the initial priors $\text{ex}[\mathcal{Q}_0]$, so we can discard them once the posteriors $\mathcal{Q}_1$ are learned in the first task.

The overall time complexity is dominated by the $O(v)$ variational inference in Algorithm 1, used as a subroutine in line 3. Compared to variational inference, the $O(1)$ preferred model generation via convex

combination in Algorithm 2 in line 7 is negligible. Therefore, the overall time complexity for $n$ tasks is $O(nv)$, regardless of preferred model generation. Moreover, as the memory complexity at each task is contributed by $O(1)$ memorization of posteriors by Algorithm 1, the total memory complexity is $O(n)$. Some of these posteriors will be discarded, as discussed in Section 4.1. Therefore, in the amortized case, Algorithm 3 ensures **sublinear buffer growth**. If, at some point of the continual learning process, all newly learned posteriors are within the distant threshold to some cached posterior, the buffer will stop growing and the total memory cost becomes constant.

The following proposition ensures that IBCL locates the user-preferred Pareto-optimal model with high probability.

**Proposition 4.2** (Probabilistic Pareto-optimality)**.** *Pick any $\alpha \in [0,1]$. The Pareto-optimal parameter $\theta^\star_{\bar{w}}$, i.e., the ground-truth parameter for $p_{\bar{w}}$, belongs to $\Theta^\alpha_{\bar{w}}$ with probability at least $1-\alpha$ under distribution $\hat{q}_{\bar{w}}$. In formulas,* $\mathrm{Pr}_{\theta^\star_{\bar{w}} \sim \hat{q}_{\bar{w}}}[\theta^\star_{\bar{w}} \in \Theta^\alpha_{\bar{w}}] \geq 1-\alpha$.

Proposition 4.2 gives us a $(1-\alpha)$-guarantee in obtaining Pareto-optimal models for given task trade-off preferences. In other words, the Pareto-optimal parameter $\theta^\star_{\bar{w}}$ is guaranteed to belong to the Highest Density Region $\Theta^\alpha_{\bar{w}}$ that we build, with high probability. Our algorithm does not find the parameter $\theta^\star_{\bar{w}}$ itself, but instead the narrowest region $\Theta^\alpha_{\bar{w}}$ that contains it with high probability. In spirit, this result is very similar to what conformal prediction does (for predicted outputs, rather than parameters of interest (Angelopoulos & Bates, 2021)). Consequently, the IBCL algorithm enjoys the **probabilistic Pareto-optimality** targeted by our main problem. Please refer to Appendix B for the proof.

### 4.4    Reinforcement Learning using IBCL

Although the focus of this paper is classification tasks, we also outline a solution to apply IBCL to reinforcement learning. Here, the system is formalized as an MDP $\mathcal{M} = (\mathcal{S}, \mathcal{A}, f, r, \gamma)$, where $\mathcal{S}$ and $\mathcal{A}$ are state space and action space, respectively, $f(s_{t+1}|s_t, a_t)$ is the Markovian transition probability from a previous state $s_t \in \mathcal{S}$, an action $a_t \in \mathcal{A}$ to a next state $s_{t+1} \in \mathcal{S}$, $r : \mathcal{S} \times \mathcal{A} \times \mathcal{S} \to \mathbb{R}$ is the reward function, and $\gamma \in [0,1]$ is the discount factor when computing cumulative reward. Generally, in a multitask or continual learning setting, every task $i$ has a task-specific MDP $\mathcal{M}_i = (\mathcal{S}_i, \mathcal{A}_i, f_i, r_i, \gamma_i)$.

For reinforcement learning, the parameter $\theta$ parameterizes a control policy $\pi_\theta : \mathcal{S} \to \mathcal{A}$, where $\mathcal{S}$ and $\mathcal{A}$ are state space and action space, respectively. A parameter distribution $q$ is a Gaussian over $\theta$. Like classification, this distribution can also be updated from a prior by observed data. The data will be produced by trajectories explored. Specifically, we build Imprecise Bayesian Continual Reinforcement Learning (IBCRL) on top of Bayesian policy gradient (Ghavamzadeh & Engel, 2006). In particular, Bayesian policy gradient is a reinforcement learning version of variational inference that replaces the log likelihood with the cumulative rewards of sampled trajectories. We write down the pseudocode of this subroutine as in Algorithm 4. Here, in every epoch, from line 4 to 10, cumulative rewards are collected from exploring in the environment (MDP), and these collected data is used to update the policy distribution at line 11. We refer to the original paper (Ghavamzadeh & Engel, 2006) for further details.

Then, if we replace the variational inference on given labeled data (as in Algorithm 1) with Bayesian policy gradient in an MDP environment, we obtain the reinforcement learning version of IBCL, denoted as Incremental Bayesian Continual Reinforcement Learning (IBCRL).

## 5    Experiments

### 5.1    Setup

**Baselines.** Although there are many baseline methods for CL, only a few baselines for CLuST exist. The following CLuST baselines are selected for comparison.

1. **Convex Combination of Deterministic Models.** This is the deterministic version of our approach. It trains one deterministic model per task and combine the model weights using the preference vectors.

---

**Algorithm 4** Bayesian Policy Update (Ghavamzadeh & Engel, 2006)

---

1: **Input:** Prior policy distribution $q$, MDP $\mathcal{M} = (\mathcal{S}, \mathcal{A}, f, r, \gamma)$, number of epochs $e$, size of trajectory cache $\tau$, and trajectory (episode) length $T$
2: **Output:** Updated policy distribution $q$
3: **for** epoch in $1, \ldots, e$ **do**
4:     Sample policy $\pi \sim q$
5:     Reward cache $\mathcal{R} \leftarrow \emptyset$
6:     **for** trajectory in $1, \ldots \tau$ **do**
7:         Sample trajectory $s_0, s_1, \ldots, s_T$ using $\pi$ in $\mathcal{M}$
8:         Compute cumulative reward $R \leftarrow \sum_{t=1}^{T} \gamma^t r(s_{t-1}, a_t, s_t)$
9:         $\mathcal{R} \leftarrow \mathcal{R} \cup \{R\}$
10:     **end for**
11:     $q \leftarrow \mathsf{bayesian\_policy\_gradient}(q, \mathcal{R})$
12: **end for**

---

2. **Rehearsal-based Deterministic Models.** This is the state-of-the-art technique for CLuST (Lin et al., 2019). These methods memorize a subset of training data for every task encountered. Task preferences are then given as weights to regularize the loss on each task's memorized data. We choose (i) GEM (Lopez-Paz & Ranzato, 2017), (ii) A-GEM (Chaudhry et al., 2018), (iii) DER, and (iv) DER++ (Buzzega et al., 2020) as baselines.

3. **Rehearsal-based Bayesian Models.** We also compare IBCL with a Bayesian technique, VCL (Nguyen et al., 2018). We equip VCL with episodic memory to make it rehearsal-based and to be able to specify a preference, an approach that has been used in (Servia-Rodriguez et al., 2021). We name this baseline VCL + rehearsal.

4. **Prompt-based.** *Prompt-based CL has never been used for CLuST and, therefore, is not state-of-the-art.* Still, they are considered efficient modern CL techniques. Therefore, we attempted to specify preferences in L2P (Wang et al., 2022), a prompt-based method, by training a learnable prompt prefix per task and using a preference-weighted sum of the prompts at inference time.

**Datasets.** We experiment on four standard continual learning benchmarks, including three image classification and one NLP, as follows.

1. 5 tasks in 20 News Group (Lang, 1995) (news related to computers vs. not related to computers).

2. 10 tasks in Split CIFAR-100 (Zenke et al., 2017) (animals vs. non-animals),

3. 10 tasks in Tiny ImageNet (Le & Yang, 2015) (animals vs. non-animals), and

4. 15 tasks in CelebA (Liu et al., 2015) (with vs. without attributes).

The features are first extracted by ResNet-18 (He et al., 2016) for the first three image benchmarks. For 20 News Group, features are extracted by TF-IDF (Aizawa, 2003). For each benchmark, all tasks share the same input and label space. There is no task id at training or inference time, so the algorithm does not know which task each data point comes from. Therefore, all experiments are domain-incremental according to Van de Ven & Tolias (2019). Still, tasks arrive one-by-one in sequential temporal orders to indicate task boundaries.

**Evaluation metrics.** To evaluate how well a model addresses preferences, we randomly generate $n_{\text{prefs}}$ preferences per task, except for task 1, whose preference is always a scalar 1. Formally, at each task $i > 1$, we have preferences $\bar{w}_i^k = (w_{i1}^k, \ldots, w_{ii}^k)$ for $k = 1, \ldots n_{\text{prefs}}$.

Like all continual learning evaluations, after training on a task $i$, we first evaluate the accuracy $acc_{ij}$ of the current model on the testing sets of all tasks $j = 1, \ldots, i$ encountered so far. To do so, the method (a baseline or IBCL) computes an $i$-dimensional accuracy vector for each preference. That is, we have

$$a\bar{c}c_i^k = (acc_{i1}^k, \ldots, acc_{ii}^k), \text{ for } k = 1, \ldots, n_{\text{prefs}}. \tag{7}$$

Then, to compute the accuracy $acc_{ij}$ that takes account of all preferences, we do a weighted sum of each $acc_{ij}^k$ for all $k$. Since the preference indicates how important a task is considered, it is computed as

$$acc_{ij} = \frac{1}{\sum_{k=1}^{n_{\text{prefs}}} w_{ij}^k} \sum_{k=1}^{n_{\text{prefs}}} w_{ij}^k acc_{ij}^k. \tag{8}$$

For example, at task $i = 2$, suppose we have $n_{\text{prefs}} = 3$, with preferences $(0.5, 0.5)$, $(0.1, 0.9)$ and $(0.8, 0.2)$. The method evaluates on the testing data of task 1 with accuracies $a$, $b$ and $c$, respectively for the 3 preferences. We therefore have $acc_{21} = \frac{1}{0.5+0.1+0.8}(0.5a + 0.1b + 0.8c)$.

In the experiments, we set $n_{\text{prefs}} = 10$. After obtaining $acc_{ij}$, we use state-of-the-art continual learning metrics to evaluate performance of a task $i$ with

1. Average per task accuracy: $\frac{1}{i} \sum_{j=1}^{i} acc_{ij}$, and
2. Peak per task accuracy: $\max_{k \geq i} acc_{ki}$.

Starting from task $i = 2$, resistance to catastrophic forgetting is evaluated by

3. Backward transfer (Díaz-Rodríguez et al., 2018): $\frac{1}{i-1} \sum_{j=1}^{i-1}(acc_{ij} - acc_{i-1,j})$, with a more positive value indicating higher resistance, and a more negative value indicating higher forgetting.

**System.** Experiments are run on Intel(R) Core(TM) i7-8550U CPU @ 1.80GHz.

Detailed experiment configurations can be found in Appendix C.

### 5.2   Main Results

#### 5.2.1   Performance

In the main experiments, we learn one parameter posterior at every task, i.e. $m_k = 1$, with significance level $\alpha = 0.01$ and threshold $d = 0.012$. These hyperparameters are selected on the basis of ablation studies, which are later presented in Section 5.3.

We present the three metrics (average per task accuracy, peak per task accuracy, and backward transfer) on the four datasets. The results of 20 News Group and Split CIFAR-100 are illustrated in Figure 4, and CelebA and Tiny ImageNet in Figure 5. Our results support the claim that IBCL not only achieves high performance by probabilistic Pareto-optimality, but is also efficient with zero-shot generation of models and exhibits constant memory overheads.

From Figures 4 and 5, we can see that IBCL in general generates the model with top performance (high accuracy) in all cases, while eventually converging to no catastrophic forgetting (near zero or positive backward transfer at the last task). This is due to the probabilistic Pareto-optimality guarantee. Statistically, IBCL improves on baselines by at most 44% on average per task accuracy, and by 45% on peak per task accuracy (compared to convex combination of deterministic models in 20 News Group). So far, to our knowledge, there is no discussion on how to specify a task trade-off preference in prompt-based continual learning, and we only make an attempt for L2P, which generally works poorly. The reason for such poor performance of L2P is that we are only modifying the generated prefix embeddings to adapt to CLuST. This is only an attempt under the assumption that we do not have access to fine-tune or train the underlying large model. To provide a fairer comparison, one possible way is to directly modify the large model itself. For example, it could be augmented to a hypernetwork that accepts preferences as additional inputs. These modifications are beyond this paper and can serve as a potential future research direction.

As illustrated in the figures, IBCL has a slightly negative backward transfer at first, but then this value converges to near-zero or positive. This shows that although IBCL may slightly forget the knowledge learned from the first task in the second task, it steadily retains knowledge afterward.

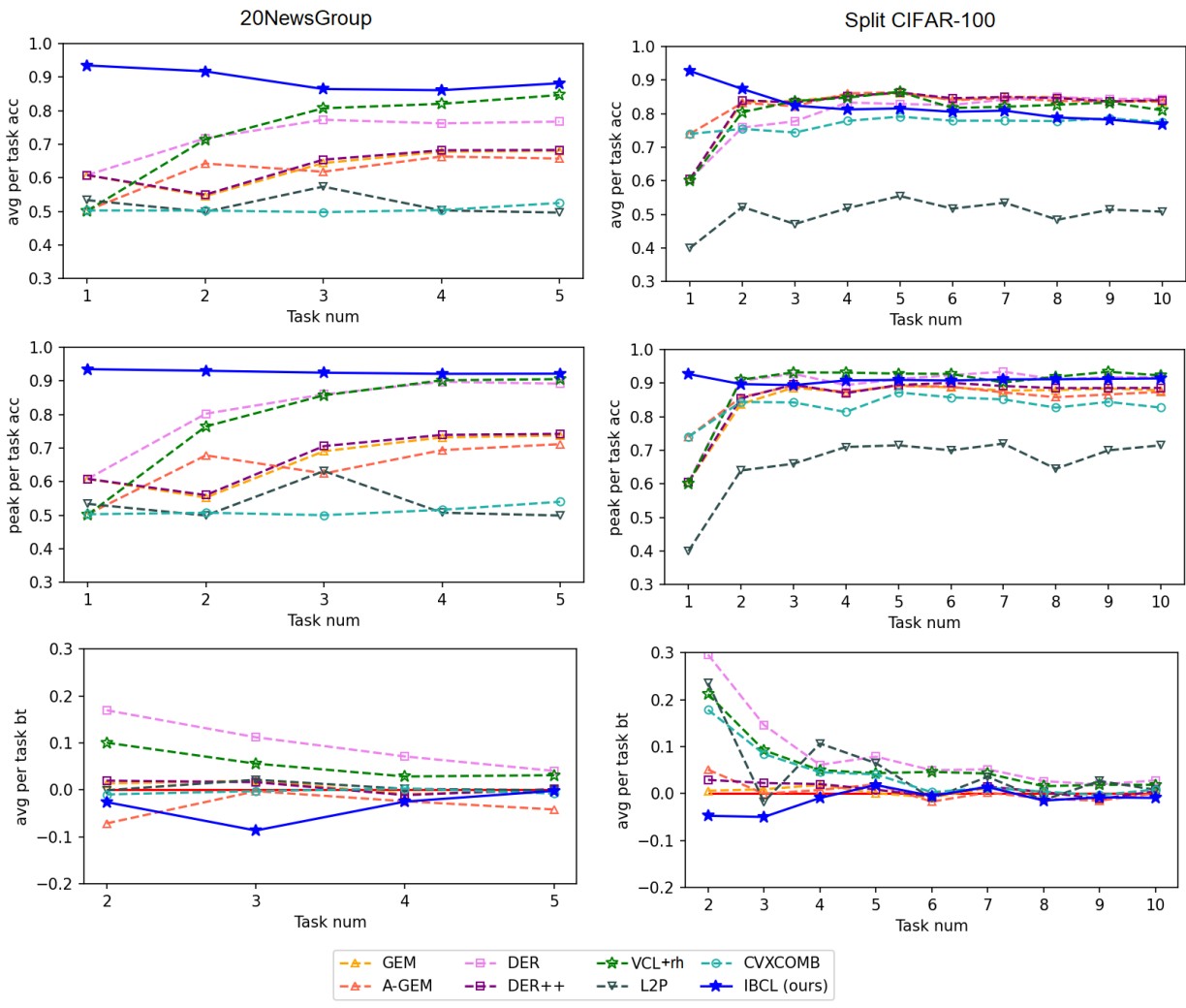

Figure 4: Results of 20 News Group (left column) and Split CIFAR-100 (right column).

Although some baselines, such as VCL + rehearsal and DER, have backward transfer higher than IBCL's in the first few tasks, IBCL eventually reaches a near-zero to positive backward transfer value. This happens at the 5th task of 20 News Group, 5th task of Split CIFAR-100, 3rd task of Tiny ImageNet, and 10th task of CelebA.

### 5.2.2 Training Time Overhead

We measure training overhead in terms of # of batch updates required at a task in Table 1. Here, $n_i$: # of training data points at task $i$, $n_{\text{prefs}}$: # of preferences per task, $n_{\text{mem}}$: # of data points memorized per task in rehearsal, $n_{\text{priors}}$: # of priors in IBCL, which is 1 in main experiments, $e$: # of epochs and $b$: batch size. Notice that the overhead of rehearsal-based methods is proportional to $n_{\text{prefs}}$, which is potentially a large number.

Table 1 shows the training overhead comparison measured in number of batch updates per task. We can see how IBCL's overhead is independent of the number of preferences $n_{\text{prefs}}$ because it only requires training for the FGCS but not for the preferred models. From this table, we see that in terms of batch updates, IBCL costs at least 6.3% as the rehearsal baselines (Split CIFAR-100) and at most 9.6% (CelebA). **Consequently,**

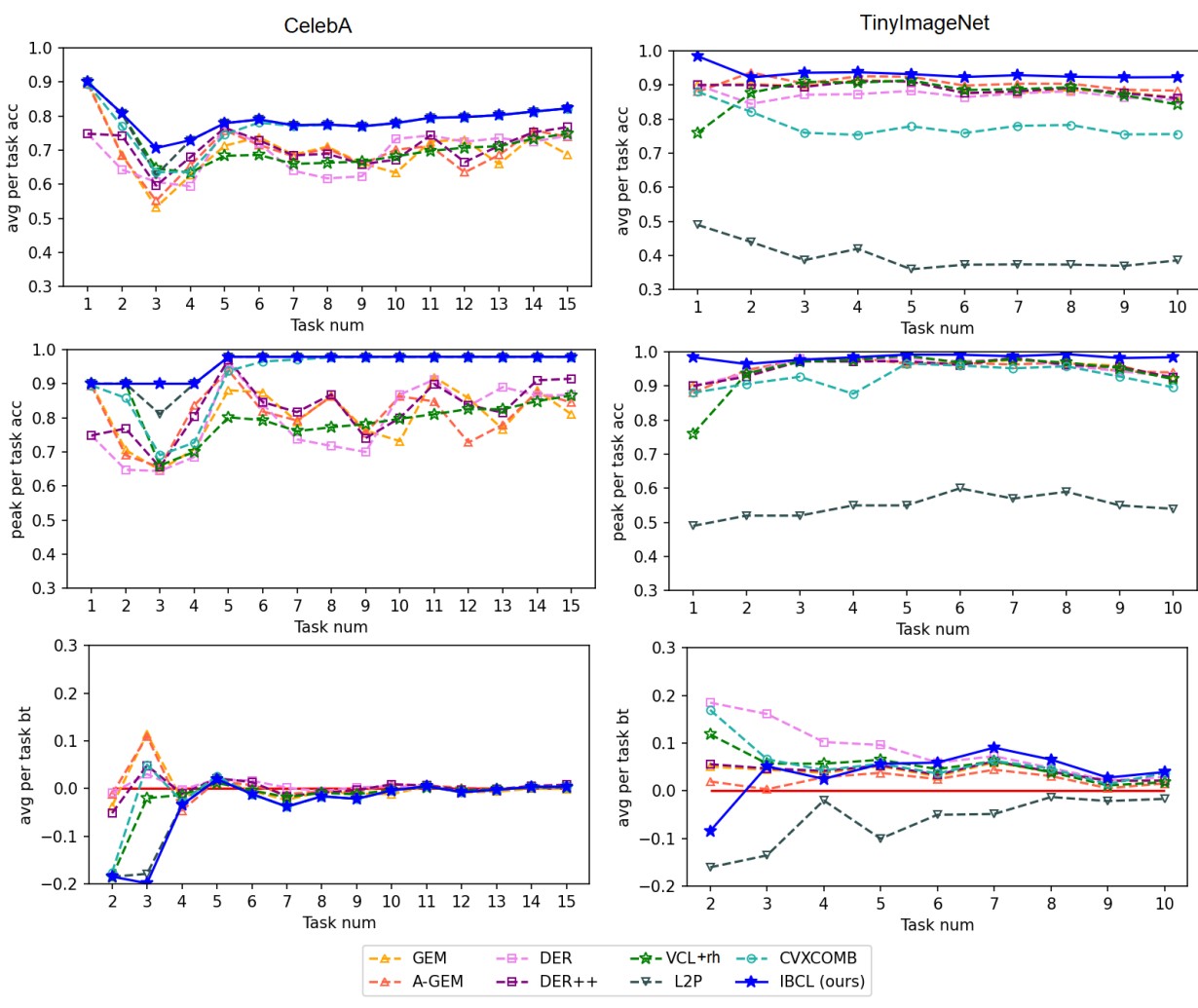

Figure 5: Results of CelebA (left column) and Tiny ImageNet (right column).

Table 1: Training overhead comparison, with hyperparameters setup in Appendix C.

| | # batch updates at task $i$ | # batch updates at last task | | | |
| --- | --- | --- | --- | --- | --- |
| | | CelebA | CIFAR100 | TImgNet | 20News |
| Convex Comb | $n_{\mathrm{prefs}} \times n_i \times e/b$ | 95384 | 12500 | 9380 | 29063 |
| GEM
A-GEM
DER
DER++
VCL + rehearsal | $n_{\mathrm{prefs}} \times (n_i + (i-1) \times n_{\mathrm{mem}}) \times e/b$ | 99747 | 19532 | 13594 | 35313 |
| L2P | $n_i \times e/b$ | **9538** | **1250** | **938** | **2907** |
| IBCL (ours) | $n_{\mathrm{priors}} \times n_i \times e/b$ | **9538** | **1250** | **938** | **2907** |

**our experiments show that IBCL is able to maintain a constant training overhead per task,**

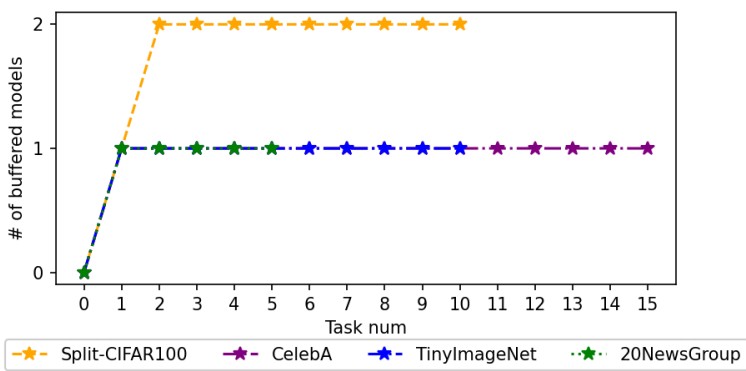

Figure 6: Number of posteriors stored along tasks.

**regardless of $n_{\mathbf{prefs}}$ while achieving high performance. Although L2P also has this constant overhead, its performance is too poor to be acceptable.**

### 5.2.3 Memory Overhead

Due to the discard threshold $d$, not all posteriors learned are cached. Figure 6 show that Split CIFAR-100 eventually converges to using 2 posterior models, while the other tasks converge to using only 1 model. This result show that IBCL is able to leverage a constant memory buffer to achieve high performance on continual learning tasks, given that the tasks are similar to each other.

We compare the overall memory cost for all methods. As described in Appendix C, each fully-connected model has 3 layers, with dimensions 512, 64, 1. Therefore, one deterministic model has $512 \times 64 + 64 \times 1 = 32897$ parameters, and one Bayesian model of Gaussian distribution has $32897 \times 2 = 65794$ parameters, with one mean and one standard deviation for each weight. Each parameter is stored as a float16, costing 2 bytes. Therefore, training one deterministic model per task and doing convex combination costs $32897 \times 2$ bytes $\times$ # tasks of memory overall.

For rehearsal-based baselines GEM, A-GEM, DER, DER++ and VCL + rehearsal, no model needs to be cached at each task. Instead, what is being cached is a replay buffer of training data. In the experiments, we use 500 data points per buffer, the same size as the original papers Lopez-Paz & Ranzato (2017); Chaudhry et al. (2018). Each data point is an extracted feature of $512 \times 2$ bytes $= 1024$ bytes. Therefore, the overall memory overhead is # data points per replay buffer $\times$ # tasks $\times$ # 1024 bytes. Although DER and DER++ cache additional information such as logits besides the replay buffer, the buffer itself is the dominant memory overhead.

For L2P, similar to the original paper (Wang et al., 2022), we have a constant prompt pool of 20 prefixes per task, with each prefix of size $5 \times 512 = 2560$ bytes. The total memory overhead is $20 \times 2560$ bytes and does not grow with the number of tasks.

At last, IBCL costs $65794 \times 2$ bytes $\times 2$ models for Split-CIFAR-100, and $65794 \times 2$ bytes $\times 1$ model for the other benchmarks. Moreover, this memory cost converges, remaining constant and not increasing along tasks.

We organize the memory overheads in Table 5.2.3. Again, although L2P saves the most memory, the use of L2P on solving CLuST problem is still merely an attempt, and it does not perform well. Consequently, among all methods, IBCL is able to not only achieve high learning performance, but also save considerable memory cost.

### 5.3 Ablation Studies

The main experiments are conducted with $\alpha = 0.01$, $d = 0.012$, and prior distributions specified in Appendix C. Here, we conduct ablation studies on these hyperparameters.

Table 2: Memory overhead comparison.

| | | Memory cost overall (KB) | | | Cost growing with |
| | 20NewsGroup | Split-CIFAR100 | CelebA | TinyImageNet | # of tasks? |
| --- | --- | --- | --- | --- | --- |
| Convex Comb | 328.97 | 657.94 | 986.91 | 657.94 | Yes |
| GEM A-GEM DER DER++ VCL + rehearsal | 2560 | 5120 | 7680 | 5120 | Yes |
| L2P | 51.2 | 51.2 | 51.2 | 51.2 | No |
| IBCL (ours) | 131.59 | 263.18 | 131.59 | 131.59 | No |

### 5.3.1 Different $d$'s

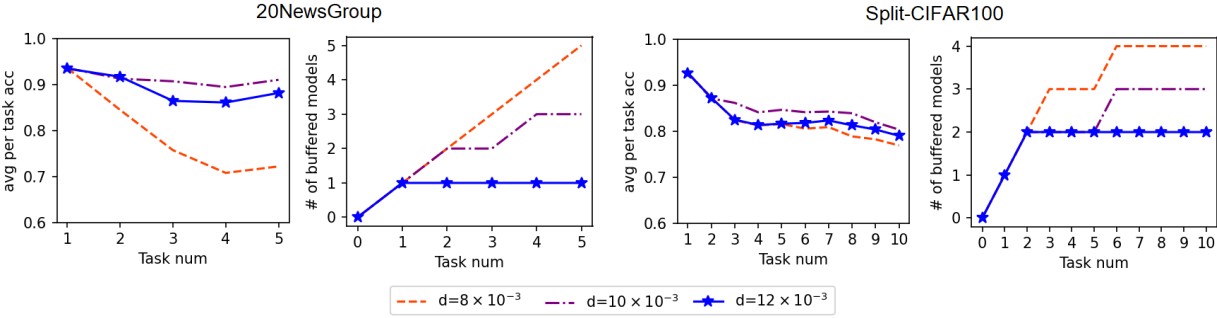

Figure 7: Different $d$'s on 20 News Group and Split CIFAR100.

Here, we evaluate the effects of choosing different thresholds $d$. We experiment on 20 News Group and Split CIFAR100. The variations include

1. $d = 12 \times 10^{-3}$, same as in the main experiments.

2. $d = 10 \times 10^{-3}$.

3. $d = 8 \times 10^{-3}$.

As $d$ increases, we are allowing more posteriors in the knowledge base to be reused. This will lead to memory efficiency at the cost of a performance drop. Figure 7 shows that when we choose $d = 12 \times 10^{-3}$ as in the main experiments, the memory cache stops growing at a certain number of tasks (task 1 for 20 News Group and task 2 for Split-CIFAR100). The learning performance slightly increases when $d$ is shrunken to $10 \times 10^{-3}$, as more distributions participate in the final evaluation. However, the performance drops when $d = 8 \times 10^{-3}$.

This observation can be explained using diversity in ensemble, as higher diversity implies improved ensemble performance (Kuncheva & Whitaker, 2003). At $d = 8 \times 10^{-3}$, more models are included in the ensemble, but they are not diverse enough from the cached models. Therefore, the errors made by the cached models and the additional models are similar, lowering the overall diversity and hence the ensemble performance. At $d = 10 \times 10^{-3}$, very-similar models are excluded, so the kept models show sufficient diversity to balance out the errors. At $d = 12 \times 10^{-3}$, more models are excluded, and too few models are kept, so there is again not enough diversity.

Overall, choosing an appropraitely large $d$, such as $d = 12 \times 10^{-3}$, is able to not only achieve sufficient performance, but also constraint the total memory cost at a constant size.

### 5.3.2    Different $\alpha$'s

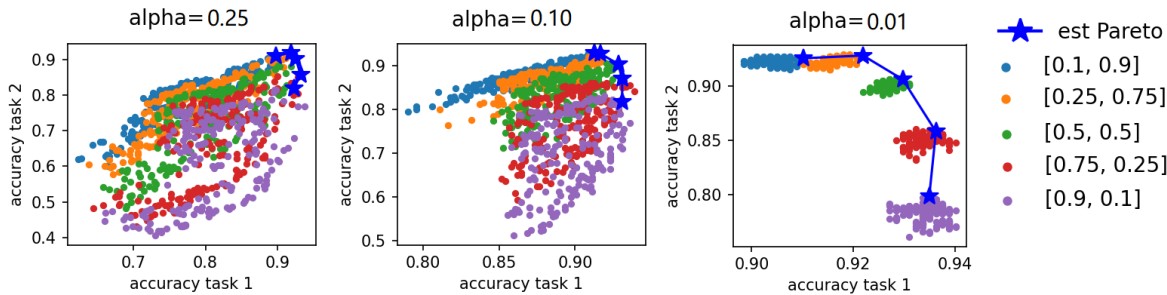

Figure 8: Different $\alpha$'s on different preferences over the first two tasks in 20 News Group.

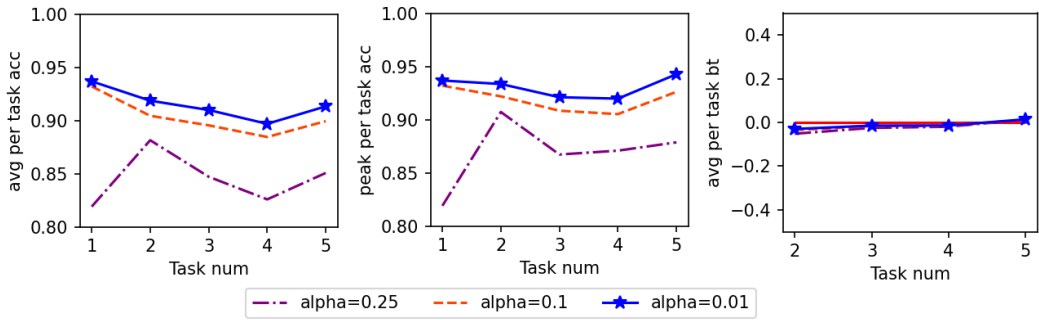

Figure 9: Different $\alpha$'s on randomly generated preferences over all tasks in 20 News Group.

Here, we evaluate the effects of choosing different significance level $\alpha$. We experiment on 20 News Group and Split CIFAR100. The variations include

1. $\alpha = 0.01$, same as the main experiments.
2. $\alpha = 0.1$.
3. $\alpha = 0.25$.

In Figure 8, we evaluate testing accuracy on three different $\alpha$'s over five different preferences (from $[0.1, 0.9]$ to $[0.9, 0.1]$) on the first two tasks of 20 News Group. For each preference, we uniformly sample 200 deterministic models from the HDR. We use the sampled model with the maximum L2 sum of the two accuracies to estimate the Pareto optimality under a preference. We can see that, as $\alpha$ approaches 0, we tend to sample closer to the Pareto front. This is because, with a smaller $\alpha$, HDRs become wider and we have a higher probability to sample Pareto-optimal models according to Proposition 4.2. For instance, when $\alpha = 0.01$, we have a probability of at least 0.99 that the Pareto-optimal solution is contained in the HDR. Figure 9 shows that the performance drops as $\alpha$ increases, because we are more likely to sample poorly performing models from the HDR.

### 5.3.3    Different Priors

As stated in Appendix C, the priors in our main experiments are decided by fine-tuning on validation sets. Here, we evaluate the effects of different priors. We experiment on 20 News Group and Split CIFAR100. First, we evaluate different sizes of prior standard deviations.

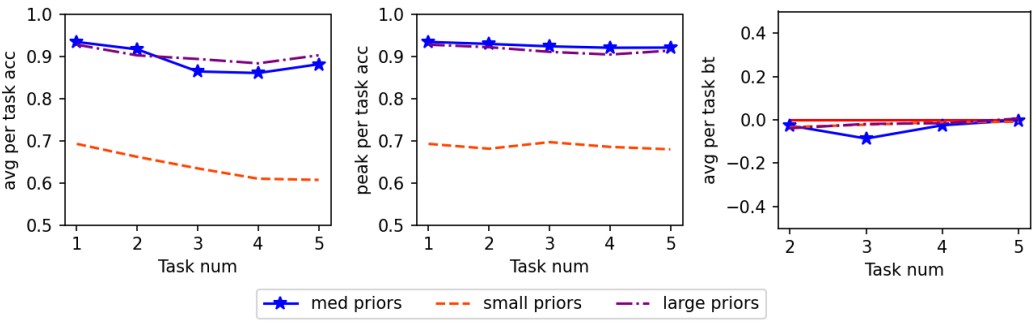

Figure 10: Different prior standard deviation sizes on randomly generated preferences over all tasks in 20 News Group.

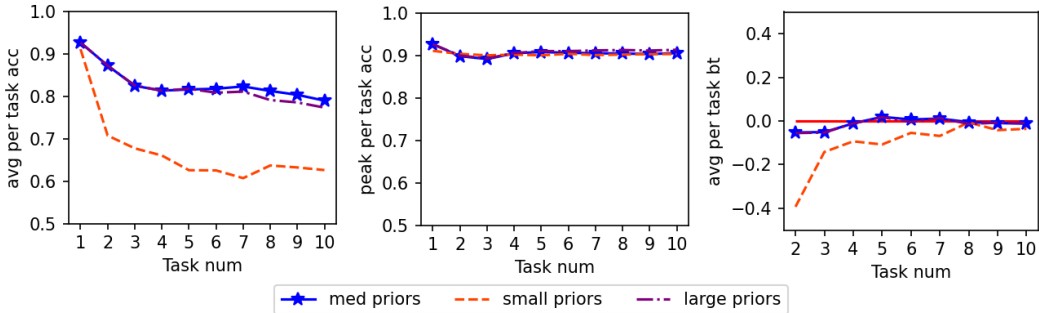

Figure 11: Different prior standard deviation sizes on randomly generated preferences over all tasks in Split CIFAR100.

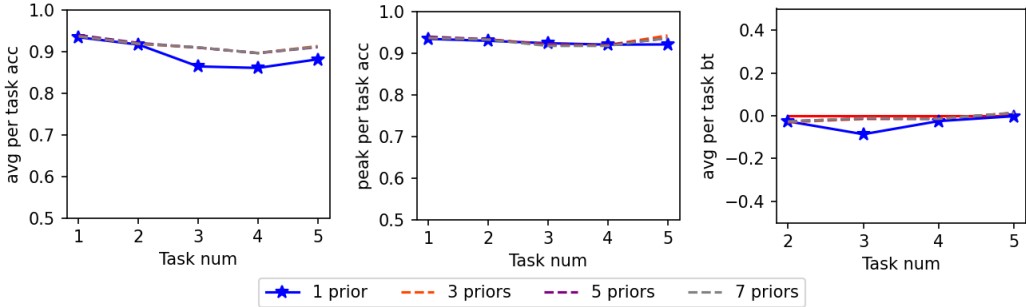

Figure 12: Different numbers of priors on randomly generated preferences over all tasks in 20 News Group.

1. Medium prior standard deviations = $\{2.5\}$, same as the main experiments.

2. Small prior standard deviations = $\{0.25\}$.

3. Large prior standard deviations = $\{25\}$.

Figure 10 and 11 show the effects of different prior standard deviation sizes on the learning performance, on 20 News Group and Split CIFAR100, respectively. We can see that in both benchmarks, small prior standard deviations lower the average and peak per task accuracy. This is because the small standard deviations lead to less variations in model parameters, leading to lower generalization. However, larger standard deviations do not necessarily mean improved performance, as the large prior standard deviations perform similarly to

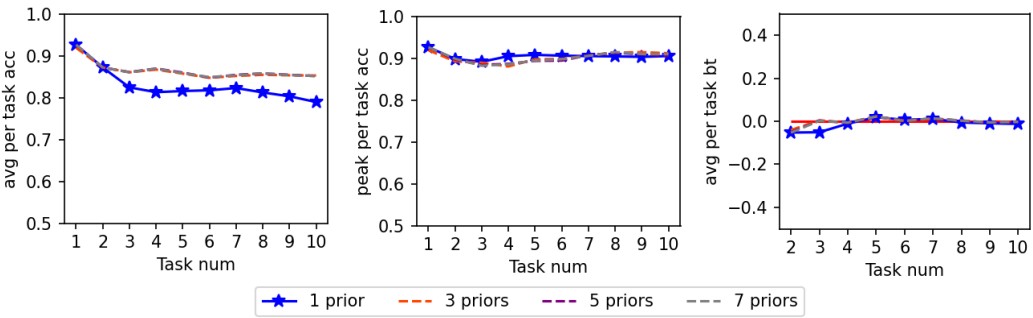

Figure 13: Different numbers of priors on randomly generated preferences over all tasks in Split CIFAR100.

the medium standard deviations in the main experiments. All backward transfers are near zero, meaning there is almost no forgetting.

We conclude that different choices of priors may lower the performance in the initial tasks. A sufficiently large prior standard deviation is needed to obtain necessary generalizability in models.

Next, we evaluate different numbers of priors. With more than one distribution per task, we balance the preference weights equally, formalized as equal $\beta$'s in Section 4.2.

1. 1 prior, standard deviation $= \{2.5\}$, same as the main experiments,
2. 3 priors, standard deviations $= \{2, 2.5, 3\}$,
3. 5 priors, standard deviations $= \{1.5, 2, 2.5, 3, 3.5\}$.
4. 7 priors, standard deviations $= \{1, 1.5, 2, 2.5, 3, 3.5, 4\}$.

As shown in Figure 12 and 13, a higher number of priors could achieve higher performance. However, more priors mean that we need to cache more models at each task, costing more memory overhead. It is a design choice to balance the trade-off between performance and memory cost. In our main experiments, we choose to use only 1 distribution per task to minimize the memory cost while maintaining a sufficiently high accuracy.

### 5.4  Additional Reinforcement Learning Experiments

In addition to the main classification tasks, we also experiment IBCRL from Section 4.4 on reinforcement learning benchmarks.

**Baselines.** We compare IBCRL to state-of-the-art multi-objective reinforcement learning (MORL) baselines.

1. **PG-MORL** (Xu et al., 2020). This method uses multi-objective gradients to update policies.
2. **PD-MORL** (Basaklar et al., 2022). This method learns a generalized policy by augmenting preferences into a Bellman equation.
3. **Hyper-MORL** (Shu et al., 2024). This method uses a hypernetwork to learn the mapping from preferences to policy parameters along side policy updates.
4. **PSL-MORL** (Liu et al., 2025). This method also uses a hypernetwork, but only mapping from preferences to a subset of policy parameters, which are implemented as a model layer.

**Benchmarks.** Like the baselines, we use standard multi-objective Mujoco (MO-Mujoco) environments (Xu et al., 2020). We pick two of these environments to run our tests.

1. MO-HalfCheetah-v2, which consists of two tasks: (1) lowering energy consumption by lowering torques and (2) improving forward speed in the $x$-direction.

2. MO-Ant-v2, which also consists of the same two tasks.

**Evaluation metrics.** Like the baselines, we use hypervolume (HV) of the learned Pareto set to evaluate the performance (Zitzler et al., 2003; Shang et al., 2020). This is the common metric used in evaluating multi-objective reinforcement learning algorithms. Specifically, a larger value indicates a solution closer to the actual Pareto front. Same as the baselines (Shu et al., 2024), we use a reference point at $(0, 0)$ and compute the average HV using nine runs over 200 preferences.

**System.** The same system is used as in the classification experiments.

Table 3: Results of hypervolume (HV) $\times 10^6$ in reinforcement learning tasks.

|  | PG-MORL | PD-MORL | Hyper-MORL | PSL-MORL | IBCRL (ours) |
|---|---|---|---|---|---|
| MO-HalfCheetah | 5.75 | **5.98** | 5.53 | 5.92 | 5.78 |
| MO-Ant | 5.79 | 7.05 | 7.49 | **8.63** | 7.67 |

The results of reinforcement learning tasks are shown in Table 5.4. We can see that IBCRL is able to maintain the same level of performance as the baselines.

However, the most important advantage of IBCL is saving training computations. Like the baseline methods, we evaluate HV on 200 preferences evenly distributed between $(1, 0)$ and $(0, 1)$. The baseline methods need to train 200 policies to do such an evaluation, while we only need to train 2 policies, for $(1, 0)$ and $(0, 1)$, respectively, and then do zero-shot convex combination to obtain the remaining 198. That is, IBCRL significantly reduces the training cost to evaluate HV.

One remark is that how to adapt IBCL to reinforcement learning is still in an elementary phase, and IBCRL is only an attempt. Improving this adaptation shall be future work.

## 6 Conclusion

We propose IBCL to tackle the CLuST problem, where models for an unbounded number of stability-plasticity trade-off preferences can be requested at each task.

**Advantages of IBCL.** The design of IBCL improves not only learning performance, but also efficiency when solving the CLuST problem, as state-of-the-art methods require retraining per preference, while IBCL only needs convex combinations. This benefit applies to various scales of models. It will be an interesting future direction to find a use case on large-scale models.

**Limitations of IBCL.** Poorly performing models can also be sampled from IBCL's HDRs. However, in practice, we can fine-tune $\alpha$ to reduce HDR to avoid poorly performing ones, as shown in ablation studies. In addition, one future research direction is to derive the preference vector $\bar{w}$ from some inputs. For example, we may learn it from an additional sequence of prompts (Wu et al., 2024). In that case, the preference vector itself might be different according to the design, including loss functions other than cross-entropy, which is currently used.

**Broader Impacts.** IBCL is potentially useful in deriving user-customized models from large multi-task models. These include large language models, recommendation systems, and other applications.

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

# Appendix A   Reason to adopt a Bayesian continual learning approach

Let $q_0(\theta)$ be our prior probability density / mass function (pdf / pmf) on the parameter $\theta \in \Theta$ at time $t = 0$. At time $t = 1$, we collect data $(\bar{x}_1, \bar{y}_1)$ related to task 1, we elicit likelihood pdf/pmf $\ell_1(\bar{x}_1, \bar{y}_1 \mid \theta)$, and we compute $q_1(\theta \mid \bar{x}_1, \bar{y}_1) \propto q_0(\theta) \times \ell_1(\bar{x}_1, \bar{y}_1 \mid \theta)$. At time $t = 2$, we collect data $(\bar{x}_2, \bar{y}_2)$ related to task 2 and we elicit likelihood pdf/pmf $\ell_2(\bar{x}_2, \bar{y}_2 \mid \theta)$. Now we have two options.

  (i) Bayesian Continual Learning (BCL): we let the prior pdf/pmf at time $t = 2$ be the posterior pdf/pmf at time $t = 1$. That is, our prior pdf/pmf is $q_1(\theta \mid \bar{x}_1, \bar{y}_1)$, and we compute $q_2(\theta \mid \bar{x}_1, \bar{y}_1, \bar{x}_2, \bar{y}_2) \propto q_1(\theta \mid \bar{x}_1, \bar{y}_1) \times \ell_2(\bar{x}_2, \bar{y}_2 \mid \theta) \propto q_0(\theta) \times \ell_1(\bar{x}_1, \bar{y}_1 \mid \theta) \times \ell_2(\bar{x}_2, \bar{y}_2 \mid \theta)$;[5]

  (ii) Bayesian Isolated Learning (BIL): we let the prior pdf/pmf at time $t = 2$ be a generic prior pdf/pmf $q_0'(\theta)$. We compute $q_2'(\theta \mid \bar{x}_2, \bar{y}_2) \propto q_0'(\theta) \times \ell_2(\bar{x}_2, \bar{y}_2 \mid \theta)$. We can even re-use the original prior, so that $q_0' = q_0$.

As we can see, in option (i) we assume that the data generating process at time $t = 2$ takes into account both tasks, while in option (ii) we posit that it only takes into account task 2. Denote by $\sigma(X)$ the sigma-algebra generated by a generic random variable $X$. Let also $Q_2$ be the probability measure whose pdf/pmf is $q_2$, and $Q_2'$ be the probability measure whose pdf/pmf is $q_2'$. Then, we have the following.

**Proposition A.1.** *Posterior probability measure $Q_2$ can be written as a $\sigma(\bar{X}_1, \bar{Y}_1, \bar{X}_2, \bar{Y}_2)$-measurable random variable taking values in $[0, 1]$, while posterior probability measure $Q_2'$ can be written as a $\sigma(\bar{X}_2, \bar{Y}_2)$-measurable random variable taking values in $[0, 1]$.*

*Proof.* Pick any $A \subset \Theta$. Then, $Q_2[A \mid \sigma(\bar{X}_1, \bar{Y}_1, \bar{X}_2, \bar{Y}_2)] = \mathbb{E}_{Q_2}[\mathbb{1}_A \mid \sigma(\bar{X}_1, \bar{Y}_1, \bar{X}_2, \bar{Y}_2)]$, a $\sigma(\bar{X}_1, \bar{Y}_1, \bar{X}_2, \bar{Y}_2)$-measurable random variable taking values in $[0, 1]$. Notice that $\mathbb{1}_A$ denotes the indicator function for set $A$. Similarly, $Q_2'[A \mid \sigma(\bar{X}_2, \bar{Y}_2)] = \mathbb{E}_{Q_2'}[\mathbb{1}_A \mid \sigma(\bar{X}_2, \bar{Y}_2)]$, a $\sigma(\bar{X}_2, \bar{Y}_2)$-measurable random variable taking values in $[0, 1]$. This is a well-known result in measure theory (Billingsley, 1986). $\square$

Of course Proposition A.1 holds for all $t \geq 2$. Recall that the sigma-algebra $\sigma(X)$ generated by a generic random variable $X$ captures the idea of information encoded in observing $X$. An immediate corollary is the following.

**Corollary A.2.** *Let $t \geq 2$. Then, if we opt for BIL, we lose all the information encoded in $\{(\bar{X}_i, \bar{Y}_i)\}_{i=1}^{t-1}$.*

In turn, if we opt for BIL, we obtain a posterior that is not measurable with respect to $\sigma(\{(\bar{X}_i, \bar{Y}_i)\}_{i=1}^{t}) \setminus \sigma(\bar{X}_t, \bar{Y}_t)$. If the true data generating process $p_t$ is a function of the previous data generating processes $p_{t'}$, $t' \leq t$, this leaves us with a worse approximation of the "true" posterior $Q^{\text{true}} \propto Q_0 \times p_t$.

The phenomenon in Corollary A.2 is commonly referred to as *catastrophic forgetting*. Continual learning literature is unanimous in labeling catastrophic forgetting as undesirable – see e.g. (Farquhar & Gal, 2019; Li et al., 2020). For this reason, in this work we adopt a BCL approach. In practice, we cannot compute the posterior pdf/pmf exactly, and we will resort to variational inference to approximate them – an approach often referred to as Variational Continual Learning (VCL) (Nguyen et al., 2018). As shown in Section 3.2, Assumption 3.2 is needed in VCL to avoid catastrophic forgetting.

## A.1   Relationship between IBCL and other BCL techniques

Like (Farquhar & Gal, 2019; Li et al., 2020), the weights in our Bayesian neural networks (BNNs) have Gaussian distribution with diagonal covariance matrix. Because IBCL is rooted in Bayesian continual learning, we can initialize IBCL with a much smaller number of parameters to solve a complex task as long as it can solve a set of simpler tasks. In addition, IBCL does not need to evaluate the importance of parameters by measures such as computing the Fisher information, which are computationally expensive and intractable in large models.

---

[5]Here we tacitly assume that the likelihoods are independent.

### A.1.1 Relationship between IBCL and MAML

In this section, we discuss the relationship between IBCL and the Model-Agnostic Meta-Learning (MAML) and Bayesian MAML (BMAML) procedures introduced in (Finn et al., 2017; Yoon et al., 2018), respectively. These are inherently different than IBCL, since the latter is a continual learning procedure, while MAML and BMAML are meta-learning algorithms. Nevertheless, given the popularity of these procedures, we feel that relating IBCL to them would be useful to draw some insights on IBCL itself.

In MAML and BMAML, a task $i$ is specified by a $n_i$-shot dataset $D_i$ that consists of a small number of training examples, e.g. observations $(x_{1_i}, y_{1_i}), \ldots, (x_{n_i}, y_{n_i})$. Tasks are sampled from a task distribution $\mathbb{T}$ such that the sampled tasks share the statistical regularity of the task distribution. In IBCL, Assumption 3.2 guarantees that the tasks $p_i$ share the statistical regularity of class $\mathcal{F}$. MAML and BMAML leverage this regularity to improve the learning efficiency of subsequent tasks.

At each meta-iteration $i$,

1. *Task-Sampling*: For both MAML and BMAML, a mini-batch $T_i$ of tasks is sampled from the task distribution $\mathbb{T}$. Each task $\tau_i \in T_i$ provides task-train and task-validation data, $D_{\tau_i}^{\mathrm{trn}}$ and $D_{\tau_i}^{\mathrm{val}}$, respectively.

2. *Inner-Update*: For MAML, the parameter of each task $\tau_i \in T_i$ is updated starting from the current generic initial parameter $\theta_0$, and then performing $n_i$ gradient descent steps on the task-train loss. For BMAML, the posterior $q(\theta_{\tau_i} \mid D_{\tau_i}^{\mathrm{trn}}, \theta_0)$ is computed, for all $\tau_i \in T_i$.

3. *Outer-Update*: For MAML, the generic initial parameter $\theta_0$ is updated by gradient descent. For BMAML, it is updated using the Chaser loss (Yoon et al., 2018, Equation (7)).

Notice how in our work $\bar{w}$ is a probability vector. This implies that if we fix a number of task $k$ and we let $\bar{w}$ be equal to $(w_1, \ldots, w_k)^\top$, then $\bar{w} \cdot \bar{p}$ can be seen as a sample from $\mathbb{T}$ such that $\mathbb{T}(p_i) = w_i$, for all $i \in \{1, \ldots, k\}$.

Here lies the main difference between IBCL and BMAML. In the latter the information provided by the tasks is used to obtain a refinement of the (parameter of the) distribution $\mathbb{T}$ on the tasks themselves. In IBCL, instead, we are interested in the optimal parameterization of the posterior distribution associated with $\bar{w} \cdot \bar{p}$. Notice also that at time $k+1$, in IBCL the support of $\mathbb{T}$ changes: it is $\{p_1, \ldots, p_{k+1}\}$, while for MAML and BMAML it stays the same.

Also, MAML and BMAML can be seen as ensemble methods, since they use different values (MAML) or different distributions (BMAML) to perform the Outer-Update and come up with a single value (MAML) or a single distributions (BMAML). Instead, IBCL keeps distributions separate via FGCS, thus capturing the ambiguity faced by the designer during the analysis.

Furthermore, we want to point out how while for BMAML the tasks $\tau_i$ are all "candidates" for the true data generating process (dgp) $p_i$, in IBCL we approximate the pdf/pmf of $p_i$ with the product $\prod_{h=1}^{i} \ell_h$ of the likelihoods up to task $i$. The idea of different candidates for the true dgp is beneficial for IBCL as well: in the future, we plan to let go of Assumption 3.2 and let each $p_i$ belong to a credal set $\mathcal{P}_i$. This would capture the epistemic uncertainty faced by the agent on the true dgp.

To summarize, IBCL is a continual learning technique whose aim is to find the correct parameterization of the posterior associated with $\bar{w} \cdot \bar{p}$. Here, $\bar{w}$ expresses the developer's preferences on the tasks. MAML and BMAML, instead, are meta-learning algorithms whose main concern is to refine the distribution $\mathbb{T}$ from which the tasks are sampled. While IBCL is able to capture the preferences of, and the ambiguity faced by, the designer, MAML and BMAML are unable to do so. On the contrary, these latter seem better suited to solve meta-learning problems. An interesting future research direction is to come up with imprecise BMAML, or IBMAML, where a credal set $\mathrm{Conv}(\{\mathbb{T}_1, \ldots, \mathbb{T}_k\})$ is used to capture the ambiguity faced by the developer in specifying the correct distribution on the possible tasks. The process of selecting one element from such credal set may lead to computational gains.

## Appendix B   Proofs of the Propositions

*Proof of Proposition 4.1.* Without loss of generality, suppose we have encountered $i = 2$ tasks so far, so the FGCS is $\mathcal{Q}_2$. Let $\text{ex}[\mathcal{Q}_1] = \{q_1^j\}_{j=1}^{m_1}$ and $\text{ex}[\mathcal{Q}_2] \setminus \text{ex}[\mathcal{Q}_1] = \{q_2^j\}_{j=1}^{m_2}$. Let $\hat{q}$ be any element of $\mathcal{Q}_2$.

Since $\mathcal{Q}_2$ is a convex set, with extreme elements $\{q_1^j\}_{j=1}^{m_1} \cup \{q_2^j\}_{j=1}^{m_2}$, there exists a probability vector $\bar{\beta} = (\beta_1^1, \ldots, \beta_1^{m_1}, \beta_2^1, \ldots, \beta_2^{m_2})^\top$ such that

$$\hat{q} = \sum_{j=1}^{m_1} \beta_1^j q_1^j + \sum_{j=1}^{m_2} \beta_2^j q_2^j. \tag{9}$$

That is, $\beta_1^j \geq 0$, $\beta_2^j \geq 0$, for all $j$, and $\sum_{j=1}^{m_1} \beta_1^j + \sum_{j=1}^{m_2} \beta_2^j = 1$. Due to the fact that every $q_1^j$ is learned by variational inference (Nguyen et al., 2018) from a prior $q_0^j$ in Algorithm 1, for each $q_1^j$, we have

$$q_1^j(\theta) \approx \frac{\ell_1(\bar{x}_1, \bar{y}_1 | \theta) q_0^j(\theta)}{\int_\Theta \ell_1(\bar{x}_1, \bar{y}_1 | \theta) q(\theta) d\theta} \propto \ell_1(\bar{x}_1, \bar{y}_1 | \theta) q_0^j(\theta) = \hat{p}_1(\bar{x}_1, \bar{y}_1 | \theta) q_0^j(\theta) \tag{10}$$

where $\ell_1$ is the likelihood at task 1, and $\hat{p}_1 \equiv \ell_1$ estimates the pdf of task 1's true data generating process $p_1$.

Recall that in Bayesian continual learning, we use the previous task's posterior as the next task's prior. Then, since every $q_2^j$ is learned by variational inference from a prior $q_1^j$, we have that

$$q_2^j(\theta) \propto \ell_2(\bar{x}_2, \bar{y}_2 | \theta) \underbrace{q_1^j(\theta)}_{\propto \ell_1(\bar{x}_1, \bar{y}_1 | \theta) q_0^j(\theta)} \propto \underbrace{\ell_2(\bar{x}_2, \bar{y}_2 | \theta) \ell_1(\bar{x}_1, \bar{y}_1 | \theta)}_{=: \hat{p}_2(\bar{x}_1, \bar{y}_1, \bar{x}_2, \bar{y}_2 | \theta)} q_0^j(\theta), \tag{11}$$

where $\hat{p}_2 := \ell_1 \times \ell_2$ estimates the pdf of task 2's true data generating process $p_2$. In general, $\hat{p}_i = \prod_{k=1}^i \ell_k$, and $\ell_k$ is the likelihood at task $k$ (Servia-Rodríguez et al., 2021). Distribution $\hat{p}_k$ estimates the pdf of the true data generating process $p_k$ of task $k$, $k \in \{1, \ldots, i\}$. Therefore, we expand on equation 9 as

$$\hat{q} = \sum_{j=1}^{m_1} \beta_1^j q_1^j + \sum_{j=1}^{m_2} \beta_2^j q_2^j \propto \hat{p}_1 \sum_{j=1}^{m_1} \beta_1^j q_0^j + \hat{p}_2 \sum_{j=1}^{m_2} \beta_2^j q_0^j. \tag{12}$$

As a consequence of the proportionality relation in equation 12, we can then find a vector $\bar{w} = (w_1 = \sum_{j=1}^{m_1} \beta_1^j, w_2 = \sum_{j=1}^{m_2} \beta_2^j)^\top$ that expresses the designer's preferences over tasks 1 and 2. In turn, we can write $\hat{q} \equiv \hat{q}_{\bar{w}}$. As we can see, then, the act of selecting a generic distribution $\hat{q} \in \mathcal{Q}_2$ is equivalent to specifying a preference vector $\bar{w}$ over tasks 1 and 2. This concludes the proof.   □

*Proof of Proposition 4.2.* For maximum generality, assume $\Theta$ is uncountable. Recall from Definition 2.2 that $\alpha$-*level Highest Density Region* $\Theta_{\bar{w}}^\alpha$ is defined as the subset of the parameter space $\Theta$ such that

$$\int_{\Theta_{\bar{w}}^\alpha} \hat{q}_{\bar{w}}(\theta) d\theta \geq 1 - \alpha \quad \text{and} \quad \int_{\Theta_{\bar{w}}^\alpha} d\theta \text{ is a minimum.}$$

We need $\int_{\Theta_{\bar{w}}^\alpha} d\theta$ to be a minimum because we want $\Theta_{\bar{w}}^\alpha$ to be the smallest possible region that gives us the desired probabilistic coverage. Equivalently, from Definition 2.3 we can write that $\Theta_{\bar{w}}^\alpha = \{\theta \in \Theta : \hat{q}_{\bar{w}}(\theta) \geq \hat{q}_{\bar{w}}^\alpha\}$, where $\hat{q}_{\bar{w}}^\alpha$ is the largest constant such that $\Pr_{\theta \sim \hat{q}_{\bar{w}}}[\theta \in \Theta_{\bar{w}}^\alpha] \geq 1 - \alpha$. Our result $\Pr_{\theta_{\bar{w}}^\star \sim \hat{q}_{\bar{w}}}[\theta_{\bar{w}}^\star \in \Theta_{\bar{w}}^\alpha] \geq 1 - \alpha$, then, comes from the fact that $\Pr_{\theta_{\bar{w}}^\star \sim \hat{q}_{\bar{w}}}[\theta_{\bar{w}}^\star \in \Theta_{\bar{w}}^\alpha] = \int_{\Theta_{\bar{w}}^\alpha} \hat{q}_{\bar{w}}(\theta) d\theta$, a consequence of a well-known equality in probability theory (Billingsley, 1986).   □

## Appendix C   Details of Experiment Configurations

We select 15 tasks from CelebA. All tasks are binary image classification on celebrity face images. Each task $i$ is to classify whether the face has an attribute such as wearing eyeglasses or having a mustache. The first 15 attributes (out of 40) in the attribute list (Liu et al., 2015) are selected for our tasks. The training, validation and testing sets are already split upon download, with 162,770, 19,867 and 19,962 images, respectively. All images are annotated with binary labels of the 15 attributes in our tasks. We use the same training, validation and testing set for all tasks, with labels being the only difference.

We select 20 classes from CIFAR100 (Krizhevsky et al., 2009) to construct 10 Split-CIFAR100 tasks (Zenke et al., 2017). Each task is a binary image classification between an animal class (label 0) and a non-animal class (label 1). The classes are (in order of tasks):

1. Label 0: aquarium fish, beaver, dolphin, flatfish, otter, ray, seal, shark, trout, whale.

2. Label 1: bicycle, bus, lawn mower, motorcycle, pickup truck, rocket, streetcar, tank, tractor, train.

That is, the first task is to classify between aquarium fish images and bicycle images, and so on. We want to show that the continual learning model incrementally gains knowledge of how to identify animals from non-animals throughout the task sequence. For each class, CIFAR100 has 500 training data points and 100 testing data points. We hold out 100 training data points for validation. Therefore, at each task we have 400 $\times$ 2 = 800 training data, 100 $\times$ 2 = 200 validation data and 100 $\times$ 2 = 200 testing data.

We also select 20 classes from TinyImageNet (Le & Yang, 2015). The setup is similar to Split-CIFAR100, with label 0 being animals and 1 being non-animals.

1. Label 0: goldfish, European fire salamander, bullfrog, tailed frog, American alligator, boa constrictor, goose, koala, king penguin, albatross.

2. Label 1: cliff, espresso, potpie, pizza, meatloaf, banana, orange, water tower, via duct, tractor.

The dataset already splits 500, 50 and 50 images for training, validation and testing per class. Therefore, each task has 1000, 100 and 100 images for training, validation and testing, respectively.

20NewsGroups (Lang, 1995) contains news report texts on 20 topics. We select 10 topics for 5 binary text classification tasks. Each task is to distinguish whether the topic is computer-related (label 0) or not computer-related (label 1), as follows.

1. Label 0: comp.graphics, comp.os.ms-windows.misc, comp.sys.ibm.pc.hardware, comp.sys.mac.hardware, comp.windows.x.

2. Label 1: misc.forsale, rec.autos, rec.motorcycles, rec.sport.baseball, rec.sport.hockey.

Each class has different number of news reports. On average, a class has 565 reports for training and 376 for testing. We then hold out 100 reports from the 565 for validation. Therefore, each binary classification task has 930, 200 and 752 data points for training, validation and testing, on average respectively.

All data points are first preprocessed by a feature extractor. For images, the feature extractor is a pre-trained ResNet18 (He et al., 2016). We input the images into the ResNet18 model and obtain its last hidden layer's activations, which has a dimension of 512. For texts, the extractor is TF-IDF (Aizawa, 2003) succeeded with PCA to reduce the dimension to 512 as well.

Each Bayesian network model is trained with evidence lower bound (ELBO) loss, with a fixed feed-forward architecture (input=512, hidden=64, output=1). The hidden layer is ReLU-activated and the output layer is sigmoid-activated. Therefore, our parameter space $\Theta$ is the set of all values that can be taken by this network's weights and biases.

Hyperparameters including the variational inference prior, learning rate, batch size and number of epcohs are tuned on validation sets. The tuning results are as follows. Here, "lr" stands for learning rate.

1. CelebA: priors $= \mathcal{N}(0, 0.25^2 I)$, lr $= 1e-3$, batch size $= 64$, epochs $= 10$.

2. Split-CIFAR100: priors $= \mathcal{N}(0, 2.5^2 I)$, lr $= 5e-4$, batch size $= 32$, epochs $= 50$.

3. TinyImageNet: priors $= \mathcal{N}(0, 2.5^2 I)$, lr $= 5e-4$, batch size $= 32$, epochs $= 30$.

4. 20NewsGroup: priors $= \mathcal{N}(0, 2.5^2 I)$, lr $= 5e-4$, batch size $= 32$, epochs $= 100$.

With the numbers above, we can compute the numerical values in Table 1.

