# OpenReview forum: "IBCL: Zero-shot Model Generation under Stability-Plasticity Trade-offs"
_TMLR — Rejected by TMLR_

### Review · Reviewer_PdJk · 2025-06-27

**Summary Of Contributions:**

This paper first formalized the Continual Learning under specific trade-offs (CLuST) problem from a Bayesian perspective, regarded model parameters as random variables, specified the stability-plasticity trade-off point through preference vectors, and provided a theoretical framework for personalized model generation. An Imprecise Bayesian Continual Learning (IBCL) algorithm is proposed. It uses the Finitely Generated Credal Set (FGCS) to store the convex hull of parameter distribution of historical tasks and realizes zero-shot model generation through convex combination, avoiding caching data and retraining of traditional methods and reducing algorithm overhead. On image classification (Split CIFAR-100, Tiny ImageNet, CelebA) and NLP (20 News Group) benchmarks, IBCL improves the average per task accuracy by 45% and the peak per task accuracy by 43% compared to the baseline, with a training overhead of only 19%-29%, while maintaining near-zero backward transfer.

**Audience:**

Yes

**Broader Impact Concerns:**

No major concerns.

**Claims And Evidence:**

Yes

**Requested Changes:**

- To strengthen the theoretical foundation, the authors should incorporate discussions on Stability-Plasticity Trade-offs. This would provide a clearer conceptual framework and better justify the methodological choices.
- It is recommended to provide a systematic overview of current research progress on the CLuST problem. A comparative analysis of existing methods would help clarify IBCL's novel contributions and its advantages over alternative approaches.
- Figure 3 shows that as the number of tasks increases, the number of elements in FGCS will also increase. I hope the authors can explain whether this has any negative impact.
- The current validation is limited to classification tasks. To better assess the generalizability of the method, experiments in reinforcement learning settings, where continual learning is often critically needed, should be conducted.

**Strengths And Weaknesses:**

**Strengths:**
- This paper formulates the problem of Continual Learning under Specific Trade-offs from a Bayesian perspective. By replacing retraining with a constant-time convex combination, the training cost of each task is constant and does not increase with the number of preferences, which solves the efficiency bottleneck of rehearsal-based learning.
- The author conducted detailed experimental evaluations of IBCL from multiple angles. The experimental results show that IBCL outperforms baseline methods in various aspects, demonstrating the effectiveness of the algorithm.

**Weaknesses:**
- The paper lacks a comprehensive review of existing research on the CLuST problem, providing only background knowledge without a thorough discussion of related work.
- The experiments are solely conducted on classification tasks, leaving the scalability of the proposed method unverified in reinforcement learning decision-making tasks, which are more aligned with real-world continual learning requirements.

---

> ### Author Response · Authors · 2025-08-09
>
> We appreciate the reviewer’s insightful comments. We have revised the submission accordingly and answered the requests as follows.
>
> - “It will be better to extend the method to reinforcement learning tasks.”
>
> In the revised manuscript, we added a new subsection to discuss how to apply IBCL to reinforcement learning (Section 4.4, page 12). We conduct the reinforcement learning experiments on standard MORL benchmarks and compare to baselines (Section 5.4, pages 20-21). The conclusion is that IBCL can maintain the same level of Pareto set hypervolumes (HV) as beeline methods, and the training overhead required for evaluating HV is significantly reduced. In Section 5.4, we also note that improving this reinforcement learning version of IBCL is a potential area for future work.
>
>
> - “The authors should incorporate discussions on Stability-Plasticity Trade-offs”.
>
> We have added a discussion of stability-plasticity trade-offs as Section 2.3 (page 5).
>
> - “It is recommended to provide a systematic overview of current research progress on the CLuST problem.”
>
> We have added a systematic literature review as Section 2.4 (page 6).
>
> - “Figure 3 shows that as the number of tasks increases, the number of elements in FGCS will also increase. I hope the authors can explain whether this has any negative impact.”
>
> The negative impact would be the memory overhead. If we always require new elements to be stored in the FGCS, the memory cache will continue to grow throughout the learning process, an undesirable cost in continual learning. To show that IBCL does not suffer from this negative impact, we reran the main experiments using only one prior per task, and a distribution discard threshold $d = 0.012$ (as opposed to three priors per task, $d = 0.002$ in the previous version). The results in Section 5.2 (pages 14-17) are updated accordingly. In the updated results, we show that the number of posteriors cached in FGCS converges to a small constant value (2 for Split CIFAR-10 and 1 for the others), while maintaining high performance.

---

> > ### Comment · Reviewer_PdJk · 2025-08-11
> >
> > I thank the authors for their detailed responses to my comments. All of my concerns and questions have been resolved.

---

### Review · Reviewer_7wAm · 2025-07-03

**Summary Of Contributions:**

This paper studies the problem of continual learning under specific tradeoffs (CLuST), where based on arbitrary user preferences, maintaining high accuracy might be more important in certain past tasks than others. According to the authors, state of the art methods tackling CLuST are rehearsal-based and require retraining on the stored per-task data, using a weighted loss based on the task preferences. To overcome the alleged need to retrain for each set of task preferences, the authors propose an algorithm, IBCL, that can adapt to any set of preferences in a zero-shot fashion, without requiring any retraining. This algorithm sequentially learns a set of posteriors per task that are used as extreme elements of a convex hull. This convex hull is then seen as the space of all possible solutions under any possible task tradeoffs, and based on a specific set of preferences, a weighted combination of those extreme elements is made to obtain the desired solution.

The main contributions of the paper are:

* formalising the CLuST as a Bayesian problem: the authors argue no prior work studied CLuST under a Bayesian lens
* basing the IBCL algorithm on finitely generated credal sets (FGCS) to have a fix training cost for any arbitrary number of preferences

**Audience:**

Yes

**Broader Impact Concerns:**

No concerns.

**Claims And Evidence:**

No

**Requested Changes:**

It is possible that I misunderstood the motivation or key point of the method. If so, I kindly ask the authors to clarify where the mis-understanding lies, and would be happy to reassess my comments. If however the comments above hold, I am concerned about the validity of the contributions made by the paper. In any case, here is a list of requested changes that I would consider necessary:

* [Critical] Report memory overhead table, in the same vein as training cost overhead is reported in Table 1
* [Critical] Report main results in settings where the memory overhead eventually stabilises over tasks, or is otherwise negligible
* [Critical] Run informative ablation experiments:
    * What happens with a single prior per task?
    * What happens if the posterior of the previous task is not reused?
    * What is the cost of having a very low $\alpha$?
* It is hard to try to see any trend in the plots for backward transfer because the axes are so zoomed out.
* In the appendix, the authors explain they did a hyper parameter search for their own method, and then reused the best parameters obtained for their own method to all other baselines? If this really so, it’s not a fair way to compare to baselines.
* Authors argue there is no need for labels, maybe clarify that task boundary indicators are however necessary during training.
* How is the peak per task accuracy reported? It seems $max_j acc_{ij}$ corresponds to the maximum accuracy after training task $i$ across all tasks $j=1,...,i$. Would a more informative metric be the opposite, i.e. $max_i acc_{ij}$ showing the best-ever attained accuracy for any given task $j$?
* Certain references can create confusion: i.e. neither Zenke et al. nor Le & Lang considered animal vs. non animal tasks in their datasets/experiments, but section 5.1 suggests so.

**Strengths And Weaknesses:**

Strengths:

* The problem is interesting and seems quite relevant from an applied perspective.
* The paper is well structured, clearly written and easy to follow.

Weaknesses:

* *The main results are based on a setting that violates basic CL desiderata*. The authors state multiple times that their method has “sublinear [memory] buffer growth”. This is in theory possible if their hyper parameter $d$ is set to a high value, but Figure 8 seems to show that for the value they use in the main experiments (2x10e-3 according to them), the memory increases linearly with the number of tasks. If this is true, the paper is written in a deceptive way, and the results violate a basic CL desideratum, namely that the memory should not increase linearly with the number of tasks. If the main results are really obtained in this setting, the authors are using one stored (actually more than one!) model per task - this is clearly not a CL setting. Or is it the case that the number of stored parameters per distribution is so small that it can be considered negligible (i.e. akin to storing a small coreset of training data per task for rehearsal)? If so, please clarify and provide numbers.
* *The experiments section is weak*: Putting aside the point above, the ablation experiments are in most cases not informative, e.g. Figures 10, 11, 12, 13, 14 show essentially no difference. Why are the most basic ablations not considered? For example, what happens if there is a single distribution per task ($m=1$)? What happens if $d$ is chosen in such a way that the memory cost stays constant after a certain number of tasks? (In my opinion, this should be the only setting considered in all experiments to be relevant from a CL perspective). What is the cost of having very low $\alpha$, i.e. does it require obtaining many more samples from the HDR until some useful solution is obtained?
* *The method is poorly motivated theoretically*: Components of the method seem to have little theoretical justification. It is not clear to me that solutions lying in the convex hull defined by all the per-task distributions will be a good, or optimal solution for downstream performance on set of preferences. If they are, why do we need multiple distributions per task and what is the cost of not having them? Since we are essentially storing a set of distributions per task, what is the rationale behind using the posteriors of the past task as priors for the new one? This would make sense to me if we were getting rid to all stored distributions of previous tasks and only keeping those of the last one (as in VCL) but this does not seem to be the case.

---

> ### Author Response · Authors · 2025-08-09
>
> We appreciate the reviewer’s thoughtful comments and have revised our paper accordingly. We rephrase the reviewer’s requests and write down our responses as follows.
>
> - “The authors need to report the memory overhead, and when there is only one prior per task, to show that the memory overhead eventually stabilizes or becomes negligible.”
>
> We have majorly edited our experiment section. In Section 5.2 (pages 14-17), we now use only one prior per task, with a discard threshold $d = 0.012$ for the main experiments. All experiments are rerun based on this new setting. We include the report on memory overhead in a new section (Section 5.2.3, pages 16-17), accompanied by a newly added Figure 2 and Table 2. The latest results show that the memory stops growing at early tasks (task 2 for Split CIFAR-100 and task 1 for the remaining benchmarks), and IBCL still maintains high performance. We also modified our ablation studies (Section 5.3, pages 17-20), where different discard thresholds are compared and discussed.
>
> - “The method is poorly motivated theoretically. It is not clear to me that solutions lying in the convex hull defined by all the per-task distributions will be a good, or optimal solution for downstream performance on set of preferences.”
>
> We thank the reviewer for giving us the opportunity to expand on this matter. Perhaps we were not clear enough in the current version of our manuscript, and will include the following comments in the final version.
>
> First, imagine that we only have one distribution per task. The elements of the ensuing convex hull are probabilities that can be written as finite convex mixtures of the per-task distributions. Interpreting every such a mixture as a distribution representing the user's preferences over the tasks (each captured by a per-task distribution), we are able to address CLuST. Notice also that this modeling choice was also put forth by researchers in Imprecise Probability. Please refer to our citations of Walley 1991, Augustin et al. 2014, and Caprio et al. 2024a in Section 2.1 (page 3).
>
> Second, in general multiple per-task distributions are not strictly needed. Our model works perfectly even with only one distribution per task. We adopt the multiple per-task distribution approach to tackle possible misspecification. That is, if the user faces model ambiguity at the beginning of the analysis, they may specify a finite number of plausible distributions, and update them as more data is gathered. The cost of not having them is to suffer from possible misspecification. If such a threat is not particularly frightening, the user can specify only one per-task distribution. Notice also that we store the updated distributions only if they are sufficiently different from the previous ones.
>
> Third, we use posteriors from previous tasks as priors for the new ones to combat catastrophic forgetting. We are not the first to introduce this idea (especially in Bayesian CL), see e.g. Kim et al. 2023 in our paper’s references. Also, we keep (some of) the previous task distributions because we want the user to be able to express their preferences on such tasks. This is indeed a new tenet, and it is useful to address CLuST problems.
>
> - “What happens if the posteriors are not reused?”
>
> Reusing posteriors as new priors is an established approach in Bayesian continual learning. We have made it clearer in Section 2.2 (pages 4-5) by citing the references (Nguyen et al., 2018; Ebrahimi et al., 2019). We have also extended Section 2.4 (page 6) to provide a thorough literature review on CLuST.
>
> - “In the appendix, the authors explain they did a hyper parameter search for their own method, and then reused the best parameters obtained for their own method to all other baselines?”
>
> This is not what we did, but we appreciate the reviewer for giving us a chance to clarify it. Our method, IBCL requires different hyperparameters from baselines, such as significance level $\alpha$ and discard threshold $d$. We search for these hyperparameters for IBCL only, by using a hold-out validation set. This is now clarified in Appendix C (pages 29-30).
>
> - “Authors argue there is no need for labels, maybe clarify that task boundary indicators are however necessary during training.”
>
> We add “Still, tasks arrive one-by-one in sequential temporal orders to indicate task boundaries” in Section 5.1 (page 13).
>
> - “How is the peak per task accuracy reported?”
>
> We thank the reviewer for catching this typo. We have edited how peak per task accuracy is computed in Section 5.1 (page 13). It is the maximal accuracy that can be achieved on a task $j$ after it is first trained on.
>
> - “Certain references can create confusion.”
>
> We removed the references to the baseline methods when describing datasets in Section 5.1 (page 13), leaving only the references to the datasets.

---

> > ### Comment · Reviewer_7wAm · 2025-08-18
> > **Answer to author's revision**
> >
> > I thank the authors for their updates addressing important revision comments. The new results showing that similar results can be obtained without linear increases in memory overhead were critically missing from the previous version of the paper.
> > Some remaining minor comments:
> > - VCL original base results were obtained without coresets, and the paper is written as if corsets were strictly necessary for this baseline (e.g. the memory overhead comparison), which can confusing/misleading to the reader - maybe rename as VCL+Coreset as in the original paper or something along those lines?
> > - I find the results in Fig 7 d=8x10-3 a bit surprising, namely that by having more models the performance is worse (20NewsGroup). These results were different in the previous version and I find the explanation not very convincing (especially when it seems that the performance is worse in later tasks, not initial tasks as the explanation implies)

---

> > > ### Comment · Reviewer_7wAm · 2025-08-19
> > > **Answer to author's revision (cntd)**
> > >
> > > Also one more comment regarding this question-answer:
> > >
> > > “In the appendix, the authors explain they did a hyper parameter search for their own method, and then reused the best parameters obtained for their own method to all other baselines?”
> > > This is not what we did, but we appreciate the reviewer for giving us a chance to clarify it. Our method, IBCL requires different hyperparameters from baselines, such as significance level  and discard threshold . We search for these hyperparameters for IBCL only, by using a hold-out validation set. This is now clarified in Appendix C (pages 29-30).
> > >
> > > If I read correctly in Appendix C, it still seems you reused important hyperparameters found for IBCL for other baselines: "For the baseline methods, we use exactly the same learning rate, batch sizes and epochs." Can you confirm whether no hyperparameter tuning was done for the baselines then? If it's the case, this gives an unfair advantage to IBCL.

---

> > > > ### Author Response · Authors · 2025-08-20
> > > >
> > > > We thank the reviewer for these thoughtful additional comments. We rephrase these comments and provide our answers below. A new version of the revised paper is submitted.
> > > >
> > > > - “VCL original base results were obtained without coresets.”
> > > >
> > > > We have renamed this baseline method to “VCL + rehearsal” throughout this paper.
> > > >
> > > > - “Why does having more models give worse performance in 20NewsGroup in Figure 7? ”
> > > >
> > > > In our previous revision, we explained that this is due to the additional models obtained at later tasks, but this may not be the most reasonable explanation, and we appreciate the reviewer for pointing this out. Here is an alternative way to explain with diversity in ensemble, as higher diversity implies higher performance [1].
> > > >
> > > > At $d = 8 \times 10^{-3}$, additional models are included in the ensemble, but they are not diverse enough from the cached models. Therefore, the errors made by the cached models and the additional models are similar, lowering the overall diversity and hence the ensemble performance. At $d = 10 \times 10^{-3}$, very-similar models are excluded, so the kept models show sufficient diversity to balance out the errors. At $d = 12 \times 10^{-3}$, more models are excluded, and too few models are kept, so there is again not enough diversity. We have modified the explanation in Section 5.3.1 on page 18.
> > > >
> > > > - “Can you confirm whether no hyperparameter tuning was done for the baselines?”
> > > >
> > > > We thank the reviewer for catching this mistake. We do not use hyperparameter tuning on the baselines. To avoid confusion, we have deleted this sentence from Appendix C.
> > > >
> > > > [1] Kuncheva, Ludmila I., and Christopher J. Whitaker. "Measures of diversity in classifier ensembles and their relationship with the ensemble accuracy." Machine learning 51.2 (2003): 181-207.

---

> > > > > ### Comment · Reviewer_7wAm · 2025-08-21
> > > > >
> > > > > I thank the authors for the updates and their answers, which have now clarified all the questions I had.

---

### Review · Reviewer_Esza · 2025-07-28

**Summary Of Contributions:**

The paper introduces Imprecise Bayesian Continual Learning (IBCL), a novel framework for Continual Learning under Specific Trade-offs (CLuST). The key innovation lies in constructing Pareto-optimal models tailored to user-defined trade-offs using a convex combination mechanism, and producing trade-off-specific models without retraining. IBCL addresses the stability-plasticity dilemma efficiently, with constant training overhead per task, and provides a principled Bayesian framework grounded in imprecise probability.

**Audience:**

Yes

**Broader Impact Concerns:**

NA.

**Claims And Evidence:**

Yes

**Requested Changes:**

Please address the weakness part and add more detailed discussion for the following questions.
1. **Extension to task-incremental learning:**
   IBCL is currently positioned within domain-incremental learning. However, the core idea of using convex combinations to construct Pareto-optimal solutions could potentially be extended to *task-incremental settings*, where trade-offs among tasks also arise (e.g., classification heads). Can the authors comment on this possible extension and its challenges?

2. **Prompt combination and fairness of L2P comparison:**
   As the preference-weighted sum of prompts in L2P may not yield preference-specific behavior. Given that L2P is rehearsal-free and relies on pretraining, while IBCL uses a different paradigm, is this comparison fair? Could L2P benefit from incorporating preference information during prompt training, and if so, how would forgetting be mitigated?

**Strengths And Weaknesses:**

### **Strengths**

1. Novel setting (CLuST): The paper clearly formulates and motivates the CLuST setting, which is relevant for real-world applications where user preferences over tasks vary.

2. Zero-shot preference-conditioned modeling: The idea of generating models for arbitrary user trade-offs without retraining is both innovative and practical.

3. Efficient knowledge updates: The use of convex combinations for knowledge updating and model generation offers an elegant and interpretable solution to balancing task preferences.

4. Comprehensive experiments: The empirical evaluation covers several baselines across different settings, including comparisons with L2P and ablation studies.

5. Clarity and structure: The paper is well-written, logically structured, and uses visual aids and bullet points effectively to guide the reader.

### **Weakness**

1. The choice of distance metric (2-Wasserstein) is justified empirically, but the impact of alternatives such as square root of JS-divergence is not explored.

2. The fairness of comparisons with L2P (especially considering pretraining and rehearsal differences) could be discussed more critically.

3. Lack of some related work discussion.
      * While Pareto set learning [1] is designed for multi-objective optimization in multi-task learning, it also supports zero-shot preference-based model generation. Could such methods be adapted to continual learning under trade-offs (CLuST)? How does IBCL fundamentally differ in its approach or assumptions?
     * In terms of tackling the stability-plasticity trade-off, how does the proposed method differ from other works that are more memory efficient [2,3]?

[1] Lin, X., Yang, Z., Zhang, X., & Zhang, Q. (2022). Pareto set learning for expensive multi-objective optimization. Advances in neural information processing systems, 35, 19231-19247.

[2] Chen, Qi, et al. "On the stability-plasticity dilemma in continual meta-learning: Theory and algorithm." Advances in Neural Information Processing Systems 36 (2023): 27414-27468.

[3] Massimo Caccia, Pau Rodriguez, Oleksiy Ostapenko, Fabrice Normandin, Min Lin, Lucas Page-Caccia, Issam Hadj Laradji, Irina Rish, Alexandre Lacoste, David Vázquez, et al. Online fast adaptation and knowledge accumulation (osaka): a new approach to continual learning. Advances in Neural Information Processing Systems, 33:16532–16545, 2020.

---

> ### Author Response · Authors · 2025-08-09
>
> We thank the reviewer for the thoughtful comments and have revised our paper accordingly. We rephrase the reviewer’s comments and write down our responses as follows.
>
> - “How to make the L2P baseline fairer in CLuST?”
>
> We agree with the reviewer that the adaptation of L2P to CLuST warrants further exploration. As we point out in Section 5, this adaptation is only an attempt. In Section 5.2 (page 14), we added a discussion on L2P. That is, the current naive adaptation is under the assumption that we do not have access to the underlying large model that generates the prefixes. To improve the adaptation, the underlying model could be modified, such as augmenting it into a hypernetwork that accepts preferences as additional input. These explorations are beyond the scope of this paper and can serve as future research topics.
>
> - “The choice of 2-Wasserstein distance is justified empirically, but alternatives, such as the square root of JS divergence, is not discussed.”
>
> We thank the reviewer for pointing out alternative metrics. We add a comment in Section 3.2 (page 8) to discuss our choice of 2-Wasserstein distance. Generally, we only need a distance metric (non-negative, symmetric, and following triangular rules) on distributions. There exist alternative metrics, such as the square root of JS divergence, but they are not necessarily efficient to compute on Gaussians. We choose the 2-Wasserstein distance because it has a closed-form expression on Gaussians that can be computed efficiently, and Gaussians are a common family of distributions in Bayesian networks.
>
> - “Can the authors comment on the potential extension to task-incremental learning?”
>
> We comment at the end of Section 3.1, page 8. This extension is trivial, as we only need to add task ids as an additional piece of input at both training and testing. Generally, domain-incremental learning is harder than task-incremental learning as the former uses strictly less information.
>
> - “Lacking some related work discussion.”
>
> We appreciate the reviewer for providing more related literature. We have added more literature review, including the ones given by the reviewer, in Sections 2.3 and 2.4 (pages 5-7). Specifically, meta learning methods are able to obtain task-specific knowledge efficiently on top of shared knowledge (primary model), but they are lacking a way to obtain models not for any particular task, but at a stability-plasticity trade-off point between tasks. This is the major motivation for follow-up research on hypernetworks, where trade-off preferences serve as an additional input. We also discuss MOBO, which aims to obtain a comprehensive posterior across all tasks, but it is generally impractical. First, it assumes data at all preference points are available, and even if so, it has to train on all of these preferences. Second, it has to update the entire comprehensive distribution when more data arrives. In contrast, IBCL only needs training on a few preferences and then generates parameters for all preferences efficiently.

---

> > ### Comment · Reviewer_Esza · 2025-08-12
> >
> > Thanks for the authors' clarification, most of my concerns have been resolved.

---

### Author Response · Authors · 2025-06-29
**Deadline for submitting revision?**

Dear Editors,

This is our first time submitting to TMLR, and we have received one review so far. May we ask

1. Will there be any more reviews?
2. What is the deadline for submitting a revision based on the reviews?

Much appreciated.

Authors

---

### Decision · Action_Editor_8hFc · 2025-10-02

**Recommendation:** Reject

**Audience:**

Yes

**Audience Explanation:**

Advances in continual learning is of interest to the community. The method introduced here, Imprecise Bayesian Continual Learning (IBCL), is a novel framework tailored to satisfy user-defined trade-off without the need of model retraining. IBCL addresses the stability-plasticity dilemma, with constant training overhead per task, and provides a principled Bayesian framework grounded in imprecise probability. These results are worth sharing with the community.

**Claims And Evidence:**

No

**Claims Explanation:**

All reviewers agree that this work tackles an interesting problem in continual learning and that the proposed solution is of interest to the community. The three reviewers had significant concerns with the original submission, which were resolved in the revised version. Examples including clarifying the theoretical motivation, including missing references, clarifying the experimental set-up, conducting additional experiments, etc.

While some discussions remained regarding the scalability of the method (ie, computational complexity and validity of the approach in the context of larger data sets), there is a more serious concern about the how the proposed method was compared to the baselines considered. The authors remained ambiguous regarding the hyperparameters that were used for these baselines. From their latest response, it seems like they did not conduct any hyperparameter optimisation, which suggests that the performance gains might not be real. The authors would have to clarify this question and possibly provide additional supporting evidence that the claims are accurate and supported experimentally.

**Resubmission Of Major Revision:**

The authors may consider submitting a major revision at a later time.